



# Exploring hail and lightning diagnostics over the Alpine-Adriatic region in a km-scale climate model

Ruoyi Cui[1], Nikolina Ban[2], Marie-Estelle Demory[1,3,4,5], and Christoph Schär[1]

[1]Institute for Atmospheric and Climate Science, ETH Zurich, Zurich, Switzerland
[2]Department of Atmospheric and Cryospheric Sciences (ACINN), University of Innsbruck, Innsbruck, Austria
[3]Wyss Academy for Nature, University of Bern, Bern, Switzerland
[4]Climate and Environmental Physics, Physics Institute, University of Bern, Bern, Switzerland
[5]Oeschger Centre for Climate Change Research, University of Bern, Bern, Switzerland

**Correspondence:** Ruoyi Cui (ruoyi.cui@env.ethz.ch)

**Abstract.** The north and south of the Alps, as well as the eastern shores of the Adriatic Sea, are hot spots of severe weather events, including hail and lightning associated with deep convection. With advancements in computing power, it has become feasible to simulate deep convection explicitly in climate models by decreasing the horizontal grid spacing to less than 4 km. These so-called kilometer-scale or convection-resolving models improve the representation of orography and reduce uncertain-

ties associated with the use of deep convection parameterizations.

In this study, we perform convection-resolving simulations for eight observed cases of severe convective events (seven with and one without observed hail) over the Alpine-Adriatic region. The simulations are performed with COSMO-crCLIM, a climate version of the Consortium for Small-scale Modeling (COSMO) regional model that runs on Graphics Processing Units (GPUs) at a horizontal grid spacing of 2.2 km. For analyzing hail and lightning, we have explored the hail growth model

(HAILCAST) and lightning potential index (LPI) diagnostics integrated with the COSMO-crCLIM model.

Comparison with available high-resolution observations reveals good performance of the model in simulating heavy precipitation, hail, and lightning. By performing a detailed analysis of three of the case studies, we identified the importance of significant meteorological factors for heavy thunderstorms that were reproduced by the model. Among these are the moist unstable boundary layer and dry mid-level air, the topographic barrier, as well as an approaching upper-level trough and cold front.

Although COSMO HAILCAST tends to underestimate the hail size on the ground, the results indicate that both HAILCAST and LPI are promising candidates for future climate research.

## 1 Introduction

Deep convective storms are ubiquitous worldwide, and severe convective events may be accompanied by hailstorms, lightning, wind gusts, and flash floods that lead to significant damage. For example, small hailstones can damage crops and vineyards,

while larger hailstones can damage roofs and buildings, resulting in considerable economic and (re-)insured losses (Punge and Kunz, 2016). The damage from individual hailstorms in Europe and the United States can exceed $1 billion USD (Púčik



et al., 2019). Therefore, it is essential to understand the spatial and temporal characteristics and associated mechanisms of such severe convective events and their potential change with the further warming of the atmosphere.

The Alpine-Adriatic region encompasses the Alps, including its southeastern extension along the Adriatic sea. It is among
the most important thunderstorm peril regions in Europe due to notable topography (Punge and Kunz, 2016) and proximity to the Mediterranean sea. Using 15 years of radar-based observations between 2002 and 2016, Nisi et al. (2016, 2018) identified enhanced frequency of hail days along the foothills of the Alps in the northern and southern pre-Alpine region, the Jura mountains, southern Germany, and the Bavarian Alps. In contrast, over the highest part of the Alps, severe hailstorms rarely occur (Punge and Kunz, 2016). Situated over the eastern shore of the Adriatic Sea, Croatia is also exposed to frequent hail
events. Using 11,000 reports from hail stations in the period of 1981-2006, Počakal et al. (2009) identified the highest averaged number of hail days over the northern region in the continental part of Croatia, which is located in between several mountains. Also, longer hail fall duration and larger hail diameters were found in the areas around the mountains compared to the flat eastern part of Croatia. It remains a challenge to understand the characteristics and mechanisms of such severe convective events over mountainous regions due to the difficulties in observing and modeling such events.

Due to the rarity, the local scale of hail events and the sparseness of stations, hailstorms are not well captured by ground-based observations. Hailpad networks are one of the options that provide information about hail size, mass, and kinetic energy, but they are only available in limited hail-prone areas (Schmid et al., 1992; Počakal et al., 2009; Jelić et al., 2020). Another option to retrieve information on hailstones is the use of weather radars. Algorithms based on radar reflectivity have successfully quantified precipitation amounts and the occurrence and properties of hail (Germann et al., 2022). With this, continuous
information about the spatial and temporal distributions of hail on a national scale can be provided, for example, over the contiguous United States (Cintineo et al., 2012) and the Alpine region (Nisi et al., 2016, 2018; Barras et al., 2021). These studies indicate that the combination of ground-based hailpads and radar-based products can provide valuable information for hail analysis on weather and climate time scales and support the evaluation of models.

Although severe convective events can cause catastrophic damage, our understanding of the associated mechanisms is still
limited due to the multiple mesoscale processes involved. The key ingredients of severe thunderstorms are conditional instability often associated with high convective available potential energy (CAPE), sufficient low-level moisture, lifting mechanisms that trigger the development of storms, and wind shear that can promote storm rotation (Houze, 2014). The combination of CAPE with other parameters, such as precipitation rate (Romps et al., 2014) or a 0-6 km deep wind-shear layer (Seeley and Romps, 2015), has been used in weather forecasting and climate research. However, the role of these ingredients in severe
storms over complex terrains is complicated (e.g. Huntrieser et al. (1997)). Kalthoff et al. (2009) used sodar, lidar, and aircraft data to investigate the 15 July 2007 storm that occurred east of the Black Forest in Germany and identified several triggering mechanisms. The high insolation during the day contributed to large latent heat fluxes that led to an accumulation of moisture in the valley, and subsequently to the mountain crest via slope winds. Strong updraughts nearly reached the level of free convection when the mesoscale convergence zone arrived and superposed with the stationary thermally induced convergence.
Trefalt et al. (2018) found that the convection initiation on 6 June 2015 in the northern Swiss Prealps was associated with strong convergence at mountain tops that propagated via cold air outflow downslope to the valley. Convergence areas commonly have





a width of about 1-2 km (Baldauf et al., 2011), so a fine and adequate representation of the convergence strength and updrafts is required to simulate a severe convective storm and investigate the driving mechanisms. At the same time, convection can be influenced and modulated by fronts, upper-level trough, cold pools, and terrain effects. Therefore, the processes involved in

each convective storm that occurs in different regions and under different synoptic situations can be very different and require specific case studies (Luo et al., 2020).

In the past decade, climate simulations at the kilometer-scale grid spacing started to emerge. The main advantages of running a model at such a high resolution are a better representation of orography and no need for a deep convection parameterization, which is often associated with large uncertainties in climate simulations (Prein et al., 2015; Leutwyler et al., 2016; Ban et al.,

2021). Such km-scale simulations lead to improved representation of the diurnal cycle of precipitation, heavy precipitation, clouds, snow, and local winds (Ban et al., 2021; Pichelli et al., 2021; Hentgen et al., 2019; Lüthi et al., 2019; Belušić et al., 2018). Still, hail and lightning are commonly not resolved or diagnosed in such models because of the complicated hail growth processes and electrification mechanisms that would make the models too expensive for climate simulations. The need to understand, predict and project hail and lightning have led to the development of diagnostic tools such as the hail

growth model HAILCAST (Adams-Selin and Ziegler, 2016) and the Lightning Potential Index (LPI) (Lynn and Yair, 2010). Such diagnostics implemented in km-scale models take advantage of a more realistic representation of convection and microphysical processes and provide information on hail and lighting without a significant increase in the computational cost of simulations. Comparison with observations shows that HAILCAST diagnostic is a good indicator of the hailstone sizes at the ground (Adams-Selin et al., 2019; Malečić et al., 2022), and LPI is highly correlated with the observed lightning flashes (Yair

et al., 2010) when the convection is well simulated. There are some models that include a more sophisticated treatment of hail and lightning processes. For instance, Meso-NH supports an explicit treatment of lightning, which represents the life cycle of the electric charges from generation to neutralization via lightning flashes, and a two-moment aerosol-could-microphysics scheme (Lac et al., 2018). Such simulations are far more expensive and currently not yet suited for simulations over climate time scales in large computational domains.

In this study, we use the Consortium for Small-scale Modeling (COSMO) model with HAILCAST and LPI diagnostics and available observations to explore severe convective events over the Alpine Adriatic region. The specific objectives of the study are:

– Evaluate the performance of the COSMO model at km-scale grid spacing in simulating hail and lightning.

– Explore the hail and lightning mechanisms and associated environments under different synoptic situations.

– Explore how these mechanisms are represented by the COSMO model.

To address the above objectives, we simulate eight cases of severe convective storms (including moderate to severe hailstorms, and one no-hail storm) over the Alpine-Adriatic region that occurred in the period from 2009 to 2018 under different synoptic conditions.

The paper is structured as follows: Section 2 describes the model configurations and diagnostics together with the available

observations and validation methods. Section 3.1 presents the eight selected cases with observed severe weather over the



Alpine-Adriatic region. Section 3.2 evaluates the performance of HAILCAST and LPI. Section 3.4 analyzes the results for four selected cases to understand the drivers of such events and how they are represented in the model. And finally, Section 4 presents a summary of the results with a discussion of the potential use of HAILCAST and LPI diagnostics in future climate simulations.

## 2 Data and methods

### 2.1 Model description

The simulations are performed with the climate version of the non-hydrostatic COSMO model (Baldauf et al., 2011). More specifically, we use COSMO-crCLIM, a version of COSMO that is able to run on hybrid CPU-GPU architectures (Leutwyler et al., 2017; Schär et al., 2020). Hereafter, we refer to COSMO-crCLIM as COSMO for simplicity. The simulations are conducted following a two-step one-way nesting approach with a horizontal grid spacing of 12 km for the first nest and 2.2 km for the second nest (Fig. 1a). The simulations are driven by the ERA5 reanalysis (Hersbach et al., 2020) with a boundary updating frequency of 1 hour. Both domains are discretized with 60 terrain-following hybrid vertical levels, where vertical spacing ranges from 20 m above the surface to about 1.2 km at the model top located at 23.5 km.

From the parameterization packages, we apply a single-moment bulk microphysics scheme with prognostic cloud water, cloud ice, graupel, rain and snow (Reinhardt and Seifert, 2006), and a radiation scheme with a $\delta$-two-stream approach (Ritter and Geleyn, 1992). For the outer 12 km domain, the Tiedtke (1989) convection scheme is turned on for shallow convection and switched off for deep and mid-level convection following Vergara-Temprado et al. (2020). For the inner 2.2 km domain, the convection parameterization scheme is switched off entirely to resolve the convection processes explicitly as far as feasible.

### 2.2 HAILCAST – hail growth model

The COSMO model is run with the HAILCAST module, a diagnostic hail growth model that predicts the size of hailstones falling to the ground. It was originally a 1D coupled cloud and hail model developed by Poolman (1992) and further improved by Brimelow et al. (2002) and Adams-Selin and Ziegler (2016). The HAILCAST version used in this study is adopted from WRF-HAILCAST (Adams-Selin and Ziegler, 2016). It is activated when two criteria are satisfied: the updraft velocity must be larger than $10\,\mathrm{m\,s^{-1}}$ within the grid column, and for more than 15 minutes. The updraft duration is accumulated if a grid column or one of its adjacent grid columns has a maximum updraft exceeding $10\,\mathrm{m\,s^{-1}}$ between the previous and current model time steps. If the two criteria are met, the vertical profile at the given model time step is passed to HAILCAST, which then calculates the evolution of five hail embryos. Two embryos of 5 and 7.5 mm in diameter are initialized at -8 °C level and three embryos of 5, 7.5, and 10 mm in diameter are initialized at -13 °C. HAILCAST is activated every 5 minutes in the inner COSMO 2.2 km domain. The output provides the maximum hailstone diameter of the five prescribed hail embryos and is stored with hourly frequency.

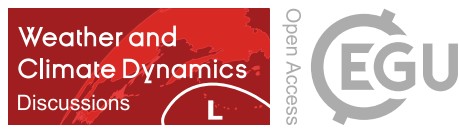

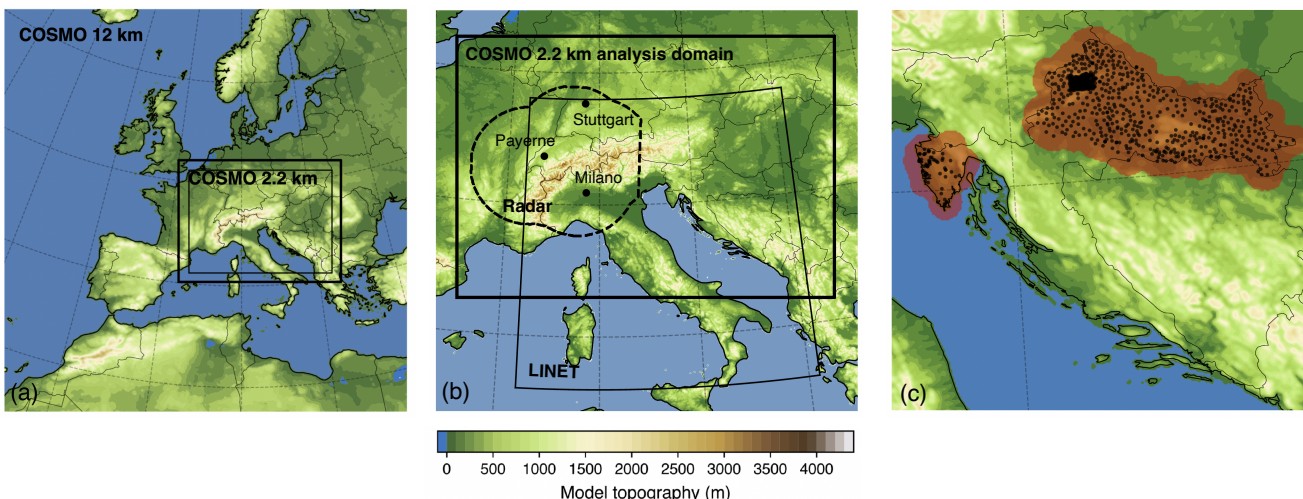

**Figure 1.** COSMO model topography, analysis domains, and observational coverage. (a) Computational domains for the simulations with 12 and 2.2 km grid spacings. The innermost box denotes the analysis domain. (b) COSMO 2.2 km analysis domain (thick solid line), LINET lightning observations (thin line), radar-based hail observations (dashed line). Black dots represent the four sounding stations used in this study: Payerne, Milano and Stuttgart. (c) Available hailpad measurements over Croatia (black dots). A dense hailpad polygon (150 hailpads aligned with a distance of around 2 km between hailpads) is located in the northwestern part of Croatia. The red-shaded area indicates the region used to evaluate hail simulated by COSMO.

## 2.3  LPI – lightning potential index

The lightning potential index (LPI, J kg$^{-1}$) is a measure of the potential for charge generation and separation that leads to lightning flashes in convective thunderstorms (Lynn and Yair, 2010; Yair et al., 2010). It considers the separation region of clouds within the main charging zone (0 to -20 °C), where the contribution of non-inductive mechanisms is the most efficient. Non-inductive mechanisms refer to the rebounding collisions between cloud ice crystals and graupel particles under the presence of





supercooled liquid water (Takahashi, 1978). We use the updated LPI version after Brisson et al. (2021):

$$LPI = f_1 f_2 \frac{1}{H_{-20^\circ C} - H_{0^\circ C}} \int\limits_{H_{0^\circ C}}^{H_{-20^\circ C}} \epsilon\, w^2 g_{(w)}\, dz \tag{1}$$

with

$$\epsilon = \frac{2(q_L q_F)^{0.5}}{q_L + q_F} \tag{2}$$

and

$$q_L = q_c + q_r, \tag{3}$$

$$q_F = q_g \left( \frac{(q_i q_g)^{0.5}}{q_i + q_g} + \frac{(q_s q_g)^{0.5}}{q_s + q_g} \right) \tag{4}$$

where $q_c, q_r, q_i, q_s$ and $q_g$ are the mixing ratios of cloud water, rain water, cloud ice, snow and graupel, respectively. $g_{(w)}$ is a boolean function equal to 1 when vertical velocity $w \geq 0.5\,\mathrm{m\,s^{-1}}$, and 0 otherwise. The dimensionless quantity $\epsilon$ scales the cloud updrafts and reaches the maximum when the vertically averaged mixing ratios of liquid ($q_L$) and combined ice ($q_F$) species are equal. Thus, the LPI is non-zero when liquid water and ice species co-exist in the grid boxes with updraft velocity above $0.5\,\mathrm{m\,s^{-1}}$, a threshold that identifies the growth phase of the thunderstorm. However, this chosen threshold generates

many LPI signals. To overcome this issue, two Boolean functions $f_1$ and $f_2$ are included to filter out weak and noisy LPI signals caused by isolated single-grid-column updrafts ($f_1$) and to filter out false LPI signals in strong orographic gravity wave clouds ($f_2$) following Brisson et al. (2021). $f_1$ is TRUE if more than 50% of grid boxes in a surrounding area of $10 \times 10\,\mathrm{km^2}$ have an updraft larger than (or equal to) a threshold $w_{max}$. The threshold $w_{max}$ is somehow arbitrary (see Brisson et al. (2021)) and depends on the grid spacing used. In our application, we have set it to $2\,\mathrm{m\,s^{-1}}$, which showed a reasonable distribution of

LPI. However, this threshold is slightly different from $1.1\,\mathrm{m\,s^{-1}}$ used by Brisson et al. (2021) with a grid spacing of $2.8\,\mathrm{km}$. $f_2$ is TRUE if a column integrated buoyancy in a surrounding area of $20 \times 20\,\mathrm{km^2}$ is larger than (or equal to) $-1500\,\mathrm{J\,kg^{-1}}$. As for $w_{max}$, this threshold is also arbitrary, but in this case, we did not do any additional test and have simply used the one recommended by Brisson et al. (2021). Thus, for more detail on these functions and choices, please see Brisson et al. (2021). LPI is calculated every 15 minutes in the COSMO 2.2 km simulations, and it is saved as an hourly maximum.

## 2.4 Observational datasets

**Precipitation observations.** The Integrated Multi-satellitE Retrievals for Global Precipitation Measurement (IMERG, Huffman et al. (2019)) dataset is used to validate the simulated precipitation. It has a spatial grid spacing of $0.1^\circ$ ($\approx 10\,\mathrm{km}$) and is available at half hourly time frequency. The IMERG data covers our entire analysis domain, including oceans that lack in-situ precipitation-measuring instruments.

In addition to IMERG, we use a gridded precipitation dataset, RhiresD, available over Switzerland only (Wüest et al., 2010). It provides daily accumulated precipitation based on a high-density rain gauge network – including 430 gauges in Switzerland. The data is available at a horizontal grid spacing of 2 km.





**Hail observations.** Simulated hail is evaluated against in-situ and remote radar observations. In-situ observations include crowd-sourced hail reports collected from the MeteoSwiss weather app (2015-present, Barras et al. (2019)) and hailpad observations retrieved from three networks located in Croatia (Fig. 1c; Počakal et al. (2009); Malečić et al. (2022)). MeteoSwiss crowd-sourced data provides information on the time, location, and size of the observed hail collected by the users of the MeteoSwiss App. The user can choose the hail size from predefined hailstone size categories: "no hail", "coffee bean", "1 Swiss Franc (CHF) coin", "5 CHF coin", and ">5 CHF coin". The size category was updated in September 2017 to include a "<coffee bean" category (to differentiate between graupel and hail) and the ">5 CHF coin" was replaced with "golf ball" and "tennis ball". More details can be found in Table 1 of Barras et al. (2019). Hailpads provide information about the number and diameters of hailstones that hit the measuring plate. The hailpad networks in Croatia include i) stations in the continental part of Croatia (590 hail stations with a mean distance of about 5.5 km between hailpads), ii) the hailpad polygon in the western part of Zagorje (150 hailpads with an equidistant spacing of 2 km), and iii) the hailpad network installed in Istra (67 hailpads) (Fig. 1c). It should be noted that hailpads observations do not report hail sizes smaller than 5 mm.

Two radar-based hail products are used to analyze hail swaths over the complex topography of the Alpine region (Fig. 1b): Probability of Hail (POH) and Maximum Expected Severe Hail Sizes (MESHS) (Nisi et al., 2016). POH is a measure of the likelihood of hail occurrence and ranges from 0% to 100%. Using crowd-sourced reports, Nisi et al. (2016) found that when POH equals or exceeds 80%, a day can be considered as hail day. The same threshold is used by Meteoswiss Swiss Hail Climatology Project, and in the current study. On the other side, MESHS estimates the largest expected hail diameter in units of centimeters, starting at 20 mm (Nisi et al., 2016). Both products are available on a spatial grid of $1\times1$ km$^2$ and every 5 minutes and cover the area of Switzerland and the surrounding area. They rely on the third-generation C-band radars (in operation since 2009) and the fourth-generation dual-polarization radars (in operation since 2012). The algorithms require information on the freezing-level height (H0) provided by the MeteoSwiss weather forecasts using COSMO. POH considers the vertical distance between the highest radar reflectivity of at least 45 dBZ and H0 (Waldvogel et al., 1979; Foote et al., 2005), while MESHS considers the vertical distance between 50 dBZ and H0 (Treloar, 1998; Joe et al., 2004). The availability of the hail data differs between the analyzed cases, so we list which hail observations are considered for each of the cases in Table 1.

**Lightning flashes.** Simulated LPI is validated against a lightning detection network (LINET) that covers large parts of Europe (Fig. 1b; Betz et al. (2009)). It has the capability to detect the total number and location of lightning strikes, where cloud-to-ground strokes, in-cloud and cloud-to-cloud discharges are included. The LINET data is provided with a spatial and temporal resolution of 3 km and 2 minutes, respectively (Jelić et al., 2021). Since the LPI output is available with hourly frequency, the LINET data is aggregated for each hour.

**Diagnostic radar reflectivity.** The COSMO model provides a diagnostic forward operator to derive an estimate of radar reflectivity. This tool will be used in some diagrams to visualize the thunderstorm development. It should be noted that this tool does not account for all aspects that contribute to radar reflectivity, for instance, it does not generate the bright band near the melting level. For these reasons, we have not used it as a validation product.





**Atmospheric soundings.** To further explore the atmospheric environments, we use 3 sounding stations (Fig. 1b) located at Payerne (Switzerland), Milano (Italy) and Stuttgart (Germany). Data is obtained from the University of Wyoming's online archive (http://weather.uwyo.edu/upperair/sounding.html.com). The soundings are available at 00 and 12 UTC.

## 2.5 Analysis methods

We evaluate daily accumulated precipitation against IMERG using an object-based verification method. Both observations and model output are first remapped to the common 12 km domain (Fig. 1b). We then use the SAL method proposed by Wernli et al. (2008) to evaluate the model performance. The SAL method compares the structure (S), amplitude (A), and location (L) of identified objects in the observations and model output. The A component is calculated as the normalized difference between the domain-averaged observations and model fields. Positive (negative) values of A indicate that the model

overestimates (underestimates) observations. The L component combines information about the spatial distance of the observed and simulated mass centers, and the error of the weighted distance between the mass centers of individual objects and the entire field. A smaller L value, i.e., closer to 0, indicates a good model representation of the observed field. The S component considers the size and shape of the objects. A positive (negative) value refers to a too widespread (peaked) modeled field. More details can be found in Wernli et al. (2008).

For each case, we have estimated whether the atmospheric instability was generated by local conditions or synoptic atmospheric processes. This classification depends on the convection adjustment scale $\tau$, which is derived using the precipitation rate $P$ (kg m$^{-2}$ s$^{-1}$) and $CAPE$ according to the following equation (Keil et al., 2013) :

$$\tau \sim \frac{CAPE}{dCAPE/dt} \sim \frac{1}{2}\frac{C_p}{L_v}\frac{\rho T_0}{g}\frac{CAPE}{P} \tag{5}$$

where in the second equation $dCAPE/dt$ is estimated from precipitation $P$. Reference values of density ($\rho = 1.292$ kg m$^{-3}$), temperature ($T_0 = 273.15$ K), specific heat of air at constant pressure ($c_p$), latent heat of vaporization ($L_v$) and acceleration due

to gravity ($g$) are taken. $\tau$ considers the timescale within which CAPE is removed by convection. Keil et al. (2013) suggests that if $\tau$ is shorter than 12 hours, the atmospheric instability is governed by the synoptic conditions, and the event is then classified as strong synoptic forcing. A larger $\tau$ ($> 12$ hours), however, indicates that the convection is driven by high local CAPE values, in which case the event is classified as weak synoptic forcing. We should note that the threshold between weak and strong synoptic forcing varies in literature (see e.g., Zimmer et al. (2011)), and should thus not be taken strictly, especially

for the cases close to it. We use the hourly domain-averaged CAPE and precipitation from ERA5 (same domain as in Fig. 2) and calculate the daily maximum $\tau$ for each case, as it must be calculated over a region large enough to smooth the variability from individual clouds. Here we just use it as an indication of prevailing conditions.

## 2.6 Predictability with adopted simulation strategy

A central element of the simulation strategy is the use of high spatial (30 km) and high temporal (hourly) ERA5 lateral boundary
conditions, with the initialization taking place at 12 UTC on the day before the event to account for the spinup of the storms. The simulation is thus guided along the reanalysis, and the predictability of our simulations is much higher than a numerical





weather prediction (NWP) forecast. The strategy is ideal to test diagnostic tools that require adequate synoptic forcing. Despite the use of ERA5 lateral boundaries, there is some inherent internal variability. To test the effect of model internal variability on our results, we have conducted a small ensemble of simulations for three events by shifting the initialization by +6 and –6 hours (Section 3.3). This procedure follows previous studies of Walser et al. (2004) and Hohenegger and Schär (2007).

## 3 Results

### 3.1 Synoptic overview of selected cases

– *23 July 2009.* Heavy hailstorms occurred over eastern and central Switzerland and caused damage to buildings amounting to around 261 million CHF in Switzerland (NCCS, 2021). The weather over Central Europe was dominated by a southwesterly flow and large temperature contrasts (Fig. 2a). Strong lifting associated with this cold front resulted in severe thunderstorms leading to several long hail swaths that can be seen in radar observations (shown later).

– *1 June 2013.* Warm and humid air transported from the northeast towards the Alps encountered cool air from the west (Fig. 2b). The event produced heavy precipitation (without hail) in a very narrow band near the foothills of the Alps and caused discharges with return periods of 10 to 30 years reported from several weather stations in central and eastern Switzerland (FEON, 2013; Grams et al., 2014). We selected this event to evaluate the ability of the model to simulate heavy precipitation without hail.

– *18 June 2013.* A low pressure system was situated over the Bay of Biscay (Fig. 2c) and brought warm and moist unstable air masses to central Europe with very large CAPE (not shown). Several localized and short-lived thunderstorm cells developed in the afternoon to the east of this system. Hailstones observed near Zurich caused massive localized damages estimated at 15 million CHF according to the building insurance of the Canton of Zürich (Gebäudeversicherung Kanton Zürich, GVZ (2013)).

– *25 June 2017.* This event is characterized by heavy precipitation which occurred south of the Alps. A thunderstorm that hit the city of Lugano in the early morning produced 81.5 mm of precipitation within an hour, which is expected over a long period of time less frequently than every 100 years (MeteoSwiss, 2017). It was the second hottest June since measurements began in 1864 (MeteoSwiss, 2017). Prior to this event, the high temperatures above 30 °C recorded in the Po Valley lasted for more than three days. A surface front was not present but a short-wave upper-level trough moving over Switzerland can be seen from the geopotential field at 500 hPa (Fig. 2d).

– *8 July 2017.* Multiple convective cells were triggered successively near the eastern edge of the Jura mountains, and then propagated and further developed in the afternoon. This event was embedded into the strong westerlies with high surface temperature ahead of a pronounced upper-level cut-off low (Fig. 2e). There was also hail in France and Southern Germany.





– **24 July 2017.** A slow-moving cut-off low passed over the northern side of the Alps. On the western side of the low, upper-level cold air advection occurred and led to an unstable environment (Fig. 2f). With the deepening of the system, low-level convergence and ascending motion initiated several thunderstorm cells to the south of the Alps, which later
shifted north-eastward with the prevailing flow.

– **17 May 2018.** Under the influence of the upper-level low over Poland (Fig. 2g), several isolated and local thunderstorms developed in the afternoon over the eastern shores of the Adriatic Sea. Hail was observed over the northern part of Istria (Croatia) according to hailpad observations. Affected by the Bise (a north-easterly wind that blows across the Swiss plateau to the north of the Alps), local rain showers developed over Switzerland without hail and lightning.

– **30 May 2018.** Scattered and widespread thunderstorms were initiated near the mountainous region of central Europe. The slow-moving storms caused significant damage across a large area. The surface pressure distribution was relatively flat (not shown), characterized by a "fair-weather" situation with weak temperature gradients over the eastern Alps. The Alpine region was affected by the southerly upper-level flow (Fig. 2h), where a trough extended over the Mediterranean and an anticyclonic curvature north of the trough axis. During the day, the southerly flow started to affect the weather in
the Alpine region. A similar situation continued the next day.

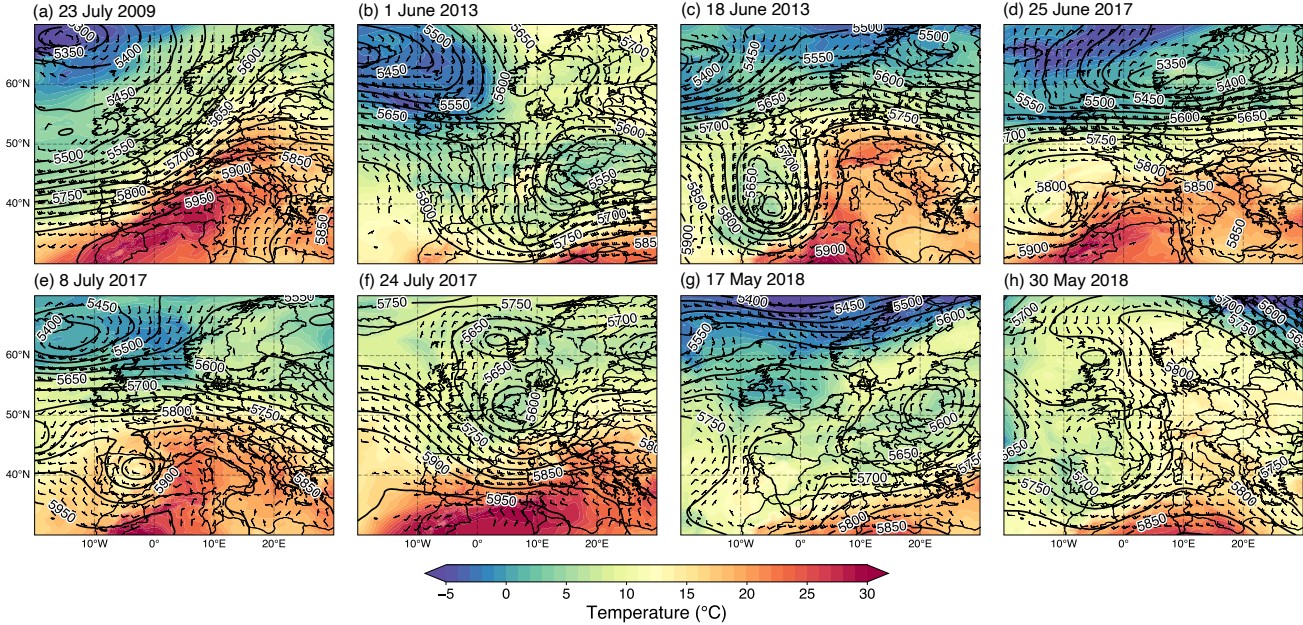

**Figure 2.** Synoptic overview of the eight case studies considered in this paper. Panels show geopotential height at 500 hPa (m, black contours), temperature at 850 hPa (°C, shaded) and wind barbs at 500 hPa obtained from ERA5 reanalysis at 12 UTC on the day when the respective case was observed.



**Table 1.** List of eight selected cases and their characteristics. The convection adjustment time $\tau$ is calculated according to the equation 2.5 and indicates cases with stronger (small $\tau$) or weaker (large $\tau$) synoptic forcing.

| Date | Description | Hail observations | $\tau$ (hours) |
|---|---|---|---|
| 23 July 2009 | Prefrontal activity north of the Alps accompanied with a south-westerly flow | radar | 10 |
| 1 June 2013 | Low-pressure system located over eastern Europe | radar | 2 |
| 18 June 2013 | Omega block with the upper-level low centered over the Iberian Peninsula | radar | 28 |
| 25 June 2017 | Passage of an upper-level trough over the Alps | radar, crowd-sourced, hailpad | 25 |
| 8 July 2017 | High temperature over central Europe leading to the development of convective cells ahead of a pronounced upper-level cut-off low | radar, crowd-sourced | 15 |
| 24 July 2017 | Two cut-off lows causing cold advection aloft over northwestern Europe | radar, hailpad | 18 |
| 17 May 2018 | A low over Poland with westerly flow over the Mediterranean | radar, hailpad | 3 |
| 30 May 2018 | Flat surface pressure field over Europe and widespread precipitation | radar, crowd-sourced | 18 |

## 3.2 Evaluation of precipitation, hail, and lightning

In this section, we assess how COSMO, with a 2.2 km grid spacing, performs in simulating precipitation, hail, and lightning. To do so, we look into the model performance with SAL diagrams (explained in Section 2.5) shown in Fig. 3, and spatial distribution of precipitation, hail, and lightning obtained from model and observations for all eight cases shown in Fig. 4-7.

Overall, the model shows a good performance in all three SAL components of precipitation in most cases (Fig. 3). The amplitude (i.e., intensity) of precipitation is overestimated for 1 June 2013 and 17 May 2018 and underestimated for 18 June 2013 and 8 July 2017. The structure component is relatively good captured for most of the cases except for 2 cases - 18 June 2013 and 25 June 2017 - for which the precipitation objects are too small and peaked compared to observations. For the case of 17 May 2018, the simulated precipitation is more scattered (Fig. 4k,o). Finally, the location component is particularly large

in two cases – 8 July 2017 and 18 June 2013. The shown bias for the 8 July 2017 case is partially due to the southerly shift of the precipitation system (Fig. 4i,m). On 18 June 2013, COSMO fails to simulate precipitation over eastern France and overestimates peak precipitation over the Black Forest (Fig. 4c,g), which results in a large location error together with the largest negative bias for amplitude and structure components.

    Daily accumulated precipitation (Fig. 4) shows very good performance of the simulations in comparison to observations.

The best performance is seen for the case of 23 July 2009 (Fig. 3 and Fig. 4a,e), characterized by stronger synoptic forcing and convection ahead of the cold front. The worst performance is seen in Fig. 3 and Fig. 4c,g for the case of 18 June 2013. The inability of the model to simulate this event properly is attributed to the local processes involved. The event is associated

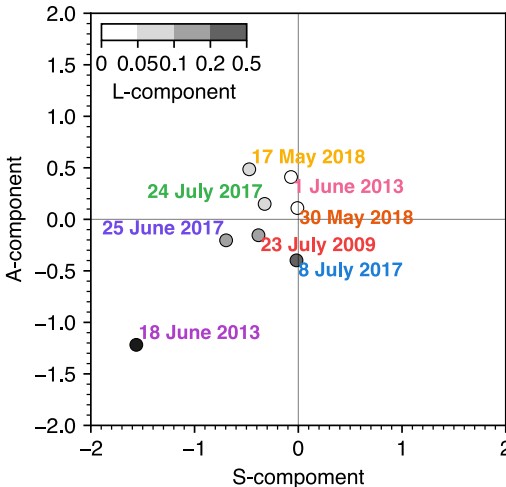

**Figure 3.** SAL diagrams of daily accumulated precipitation in COSMO simulations compared to IMERG observations over the analysis domain for all eight analyzed cases. The S, A, L components evaluate the differences in structure, amplitude, and location of the events, respectively. Values near zero signal a perfect match with observations.

with weak synoptic forcing, with the largest convective timescale of all cases (Table 1). Thus, due to its more chaotic nature, this event has small predictability. We should note, however, that the SAL components of precipitation are computed against

IMERG, which has a much coarser resolution than the model. Some of the biases can therefore be attributed to the rather smooth precipitation distribution (larger precipitation objects of lower intensity) shown by the observations (Fig. 4). It is also interesting to note that none of the used precipitation observations, neither IMERG nor RhiresD, captured the record-breaking hourly precipitation amount of 82 mm as observed at the rain gauge station in Lugano (southern Switzerland) on 25 June 2017. However, such a high precipitation intensity is simulated by the model, even though it is slightly misplaced.

To evaluate the hail produced by HAILCAST and COSMO, we first compare the model output against radar-based observations available over Switzerland and its surrounding areas (Fig. 5a-h). We first show the simulation against the POH data in terms of the hail footprint/coverage, but the comparison against MESHS data looks qualitatively similar. Overall, the model is able to reproduce hail very well compared to observations (Fig. 5). The case of 1 June 2013 with heavy precipitation but no hail over Switzerland is well reproduced, even though a very small number of grid cells produced small hail. However, the model

hail diagnostics has a tendency to overestimate the occurrence of hail in space. Among the best-simulated cases, the same as for precipitation, we can again consider the case of 23 July 2009 characterized by stronger synoptic forcing and elongated hail swaths reproduced by the model (Fig. 5a). The precipitation record-breaking event of 25 June 2017 also produced hail south of the Alps, as observed by radar and crowd-sourced reports and over the Adriatic region, including the continental part of Croatia as observed by hailpads (Fig. 5d,i,l). The widespread occurrence of hail in this case is reproduced by the model, even

though the spatial extent is overestimated. The case of 17 May 2018 is characterized by hail recorded on the Istrian penin-



**Figure 4.** Daily accumulated precipitation (mm d$^{-1}$) for all eight cases obtained from observations (first and third rows) and COSMO simulations (second and fourth rows). The IMERG observations cover the entire analysis domain, while high-resolution RhiresD gridded rain gauge observations (shown in the upper right corners) cover Switzerland only.

sula in Croatia and was well reproduced by the model. However, the model produced very light and scattered hail over both Adriatic and Alpine regions where it was not observed (Fig. 5g,n). Another case with poorer model performance is 18 June 2013, when the model overestimates the spatial extent of hail swaths, especially over the Black Forest (Fig. 5c). Overall, we can see that the performance in the simulation of hail aligns with the performance in the simulation of precipitation. However, we should note many difficulties in comparing model output with available hail observations. As described earlier, MESHS






data provides information on hailstones above 20 mm only, which can potentially lead to underestimating the hail frequency and spatial coverage. On the other side, the POH dataset only provides the probability of hail happening but does not indicate if hail has occurred or not. Last but not least, even though the hailpad network consists of many hailpad stations, many areas are not well covered and thus are prone to miss recording very localized events such as thunderstorms and hail associated with

them.

To further explore the performance of the model and HAILCAST diagnostics, we compare the simulated hail sizes against available hail size observations for different cases as listed in Table 1. Fig. 6 shows the comparison of the modeled hail against MESHS over the radar-covered area (Fig. 6a-h), crowd-sourced data over Switzerland (hailstones larger than coffee beans - 5 mm - are shown; Fig. 6i-k), and hailpad observations over the hailpad-covered area (Fig. 6a-h). When compared to MESHS

(panels a-h), the results show a large difference between the observed and simulated size distribution. We see that, while MESHS observations only show hail sizes above 20 mm, the model produces hail sizes mostly below 20 mm.

According to Barras et al. (2019), considering the 23 and 32 mm reports, MESHS tend to exceed the crowd-sourced reported hailstone size by 10-15 mm on average. Yet, compared to crowd-sourced data over Switzerland (panels i-k), the model shows a reasonable hail size distribution, although it still tends to overestimate small hailstones and underestimate large hailstones.

Moreover, compared to hailpad observations over Croatia (panels l-n), the model also shows a good distribution, particularly for the two cases of 25 June 2017 and 17 May 2018, while the case of 24 July 2017 (an event relatively well captured by the model according to the previous results) shows underestimated simulated hail sizes. The comparisons against MESHS should therefore be considered with caution. They show that there is a need to improve the radar-based hail algorithms, although comparisons with other observations show that COSMO in general tends to overestimate small hailstones and underestimate

large ones.

It is clear that the model underestimates the frequency of larger hail sizes, i.e., does not produce hail larger than 30 mm. As noted by Adams-Selin and Ziegler (2016); Adams-Selin et al. (2019), the hail size strongly depends on the initial hail size embryo – the larger the initial embryo, the larger the output hail size. However, the size also depend on the model microphysics, the strength of the updrafts that hail has to overcome to fall to the surface, and the initial temperature level. For

example, if updrafts are weaker, larger hail fall down faster and do not have enough time to grow further, while smaller hail have more time to grow but do not reach sizes above 20 mm. In a parallel study, in which the same eight cases are simulated with the WRF model (Malečić et al., 2023), larger hailstones are obtained with the WRF model than with COSMO. This result indicates that the simulated hail size strongly depends upon the model formulation.

Next, we turn our attention to the evaluation of lightning. A comparison of lightning patterns between the model and LINET

observations for each of the cases is shown in Fig. 7. In addition, to overcome difficulties related to different variables represented by observations and model (lightning flashes vs lightning potential), we calculate and display a coverage bias in the figure (red number). It is defined as the ratio of the number of gridpoints with lightning in the model and observations, respectively. Note that the coverage bias does not provide any information on the overlap of simulated and modeled lightning, but this is qualitatively assessed from the spatial representation. Overall, the model using LPI diagnostics is able to capture the

lightning patterns for each case, although it tends to slightly overestimate the spatial patterns of the signal (as for precipitation



**Figure 5.** Observed and simulated daily hail footprints for all eight cases analyzed in this study. COSMO hail footprint is shown in blue shading and compared against different observations over different regions. (a)-(h) COSMO against radar-based POH observations, shown in orange shading for the radar-covered area. A grid point with POH larger than 80% is considered a grid point with hail. (i)-(k) COSMO against crowd-sourced reports collected within Switzerland, indicated with purple dots and classified according to various categories of hail sizes. Note that after 2018, there was a change in the definition of hail sizes. (l)-(n) COSMO against hailpad measurements. Available hailpads are indicated with black dots and hailpads recording hail during the events are indicated in red for the three cases where hail occurred in Croatia.



**Figure 6.** Frequency of hail diameters obtained in observations and model simulations for all eight analyzed cases. The histograms show the frequency of hail size for each bin relative to the number of all observed/modeled hail events. (a)-(h) Maximum hail size obtained in MESHS observations (orange) and COSMO simulation (blue) over the radar domain. (i)-(k) Hail size larger than coffee beans (>5 mm) obtained from crowd-sourced observations (purple) and COSMO simulations (blue) over Switzerland. (l)-(n) Maximum hail size obtained from hailpad observations (red) and COSMO simulations (blue) over hailpad-covered areas in Croatia (refer to Fig. 1 for the domains). The black line represents the size distribution of all hailstones collected for each case. Note that the case of 1 June 2013 in (b) has no hail in the observations; the shown distribution is based on a very small number of grid points with simulated hail (see Fig. 5b).

and hail). The largest overestimation of spatial patterns, and thus the coverage bias, is found in the case of 1 June 2013, when very little lightning was observed over the Adriatic and no lightning over the Alpine region. However, the model diagnostics



produced lightning over the eastern Alps, which coincides with the area of very intense precipitation. The case of 1 June 2013
is the case without hail over the Alpine region, which was successfully reproduced by the model. Differences in representing

hail and lightning can be related to different updraft thresholds used by LPI and HAILCAST, which is lower for LPI – 0.5 m
s$^{-1}$ for LPI (Section 2.3) versus 10 m$^{-1}$ for HAILCAST (Section 2.2). The smallest coverage bias is obtained for the case of
24 July 2017 and 30 May 2018, even though there is a slight shift between the observations and the model. We should also
note that both of these cases are characterized by weaker synoptic forcing and more locally driven convection, which is well
reproduced by the model. The largest underestimation of the spatial coverage of lightning is found in the case of 25 June 2017.

345    A large part of this bias is visible over the Adriatic Sea – the area over which the model fails in reproducing precipitation as
well.

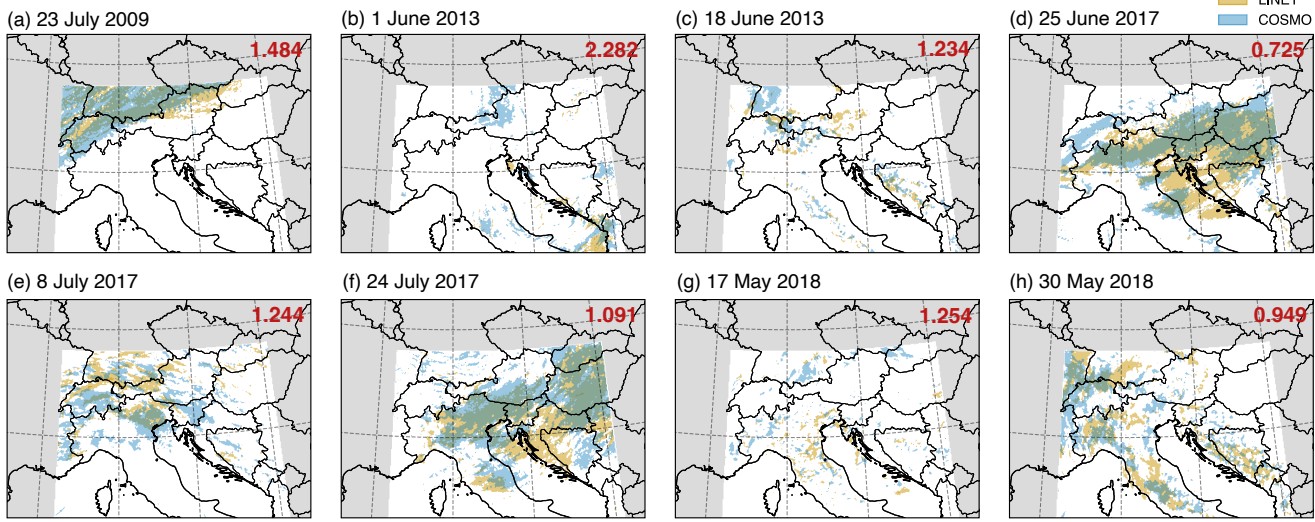

**Figure 7.** Foot prints of daily LINET lightning flashes ($>0$, orange) and COSMO LPI ($>0$ J kg$^{-1}$, blue) for all eight cases. The number in
the upper-right corner of each panel displays a coverage bias, defined as the ratio of grid points with lightning in model and observations.
Values larger and smaller than 1 indicate model overestimation and underestimation of spatial coverage, respectively.

Overall and not surprisingly, we can see that the performance of both hail and lightning diagnostics strongly depends on
simulated precipitation, since both hail and lightning diagnostics depend on the same ingredients as precipitation.

### 3.3    Assessment of model internal variability

350    A central element of the simulation strategy is the use of ERA5 lateral boundary conditions, with the initialization taking
place at 12 UTC on the day before the event to account for the spinup of the storms. The simulation is thus guided along the
reanalysis, and the predictability in our simulations is much higher than in a numerical weather prediction (NWP) forecast.
The strategy is ideal to test diagnostic tools that require adequate synoptic forcing. Despite the enhanced predictability due to





the use of ERA5 lateral boundaries, there is some remaining internal variability. To test the effect of model internal variability
on our results, we have conducted a small ensemble of simulations for three of the eight events, by shifting the initialization by
+6 and –6 hours.

The ensemble simulations are initialized at 06, 12, and 18 UTC on the day before the events occurred. Consideration is
given to the whole modeling chain with nested simulations at 12 and 2 km resolutions. Results show that even for localized
deep convective events, the predictability of precipitation and HAILCAST is overall quite high (Fig. 8). However, there are
significant differences in detail, due to the chaotic nature of the nonlinear flow evolution. For example, in the case of July 2009
(top two rows of Fig. 8), there are considerable differences in the length and location of the hail swaths. Likewise, in the case
of 18 June 2013, precipitation is simulated over the Black Forest when initialized at 12 UTC, but not when initialized at 06 and
18 UTC. Similarly, in the case of 30 May 2018, there are pronounced differences in the precipitation fields with concomitant
differences in hail. Overall, however, the internal variability is rather small and hence the simulations confirm the suitability of
the selected modeling strategy for assessing the performance of the modeling approach for case studies of severe convection.
Comparison of the cases shown in Fig. 8 suggests that synoptically-driven convective events have a higher predictability.

### 3.4 Analysis of the driving mechanisms for three specific cases

To further investigate the environmental conditions and the mechanisms that are favourable for the development of thunder-
storms over the Alpine-Adriatic region, we present a more detailed analysis of three specific cases, which affected different
areas under different synoptic situations.

#### 3.4.1 The case of 23 July 2009 – Severe thunderstorms with elongated hail swaths over Switzerland

As shown above, the case of 23 July 2009 is one of the best-simulated cases with very good performance in simulating
precipitation, hail and lightning despite overestimating the spatial extent of hail and lightning. The good performance is most
likely due to the nature of this event, which was characterized by thunderstorms ahead of a cold front and is thus classified as an
event under stronger synoptic forcing. The hail occurrence is located over areas where fronts typically become quasi-stationary
along the Alpine foothills. According to Schemm et al. (2016), up to 45% of detected hail events in north-eastern and southern
Switzerland form in this kind of pre-frontal zones.

Fig. 9a-b shows the observed and modeled Skew-T log-p thermodynamic diagram from Payerne (see Fig. 1b) at 00 and
12 UTC on 23 July 2009, which provides information on the atmospheric profile before and during the event. During the night,
a moist and stable layer below 800 hPa was located underneath a warm and dry mid-level layer. This constellation with high
observed convective inhibition (CIN) of -634 J kg$^{-1}$ (Fig. 9a) acted to suppress convection. The dry capping layer trapped
humidity in the boundary layer and accumulated energy prior to the triggering of the thunderstorms later in the afternoon.
Around noon, this profile significantly changed below 600 hPa where temperature decreased due to the advection of cold air
(Fig. 9e), while dew-point temperatures increased prominently at 800 hPa due to the moist air located ahead of the cold front.
The resulting convective cells moved northeastward (Fig. 9f) and weakened in the middle of the night. At 00 UTC on 24 July
2009, CIN was completely depleted (not shown). A comparison of the model simulation against observations at Payerne station



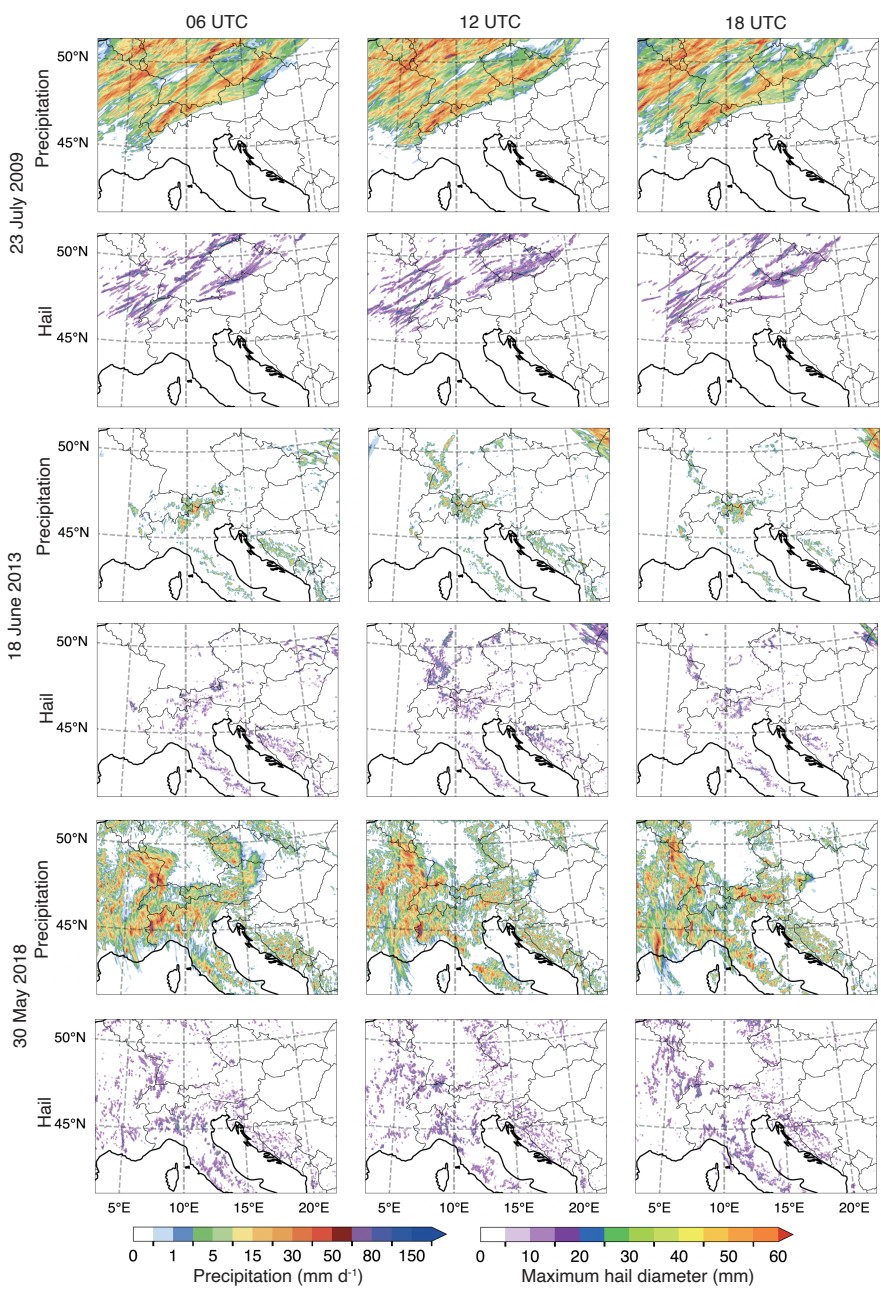

**Figure 8.** Small ensemble of simulations to explore the internal variability of the simulation strategy. Total daily precipitation and hail in simulations initiated at 06 (left), 12 (middle) and 18 (right) UTC on the day before the event occurred. All ensemble members are driven by ERA5 at the lateral boundary of the 12 km domain using hourly resolution. The results are obtained for the case of 23 July 2009 (two upper rows), 18 June 2013 (two middle rows) and 30 May 2018 (two lower rows).





reveals that COSMO reproduces this environment very well (Fig. 9b). This relatively good simulation of the storm environment
leads to a good overall performance of the model in simulating precipitation, hail and lightning during that event.

Further analysis of the case based on the model output reveals that the southwesterly flow transported warm and moist
air from the Mediterranean with an abundant water content of $35 \, \mathrm{kg} \, \mathrm{m}^{-2}$ (Fig. 9c). This warm and moist air, together with
extremely large 0-6 km bulk wind shear defined as the difference in horizontal velocity between 6 km and the surface (exceeding
$40 \, \mathrm{m} \, \mathrm{s}^{-1}$ in some areas; Fig. 9d), created favorable conditions for strong rotating updrafts. At around 05 UTC, a line of
convection developed in northeastern France and moved to the Black Forest (not shown). More detailed analysis reveals a
supercell starting near Lyon (in France), moving northeastward and gradually splitting into several elongated convection cells
seen around 12 UTC over Switzerland (Fig. 9f). The system extended for hundreds of kilometers and caused long hail swaths
and intense lightning.

### 3.4.2   The case of 25 June 2017 – A record-breaking precipitation event in Lugano

The case of 25 June 2017 is associated with record-breaking precipitation rate in Lugano during the measurement period (see
Section 3.2 above). We choose this event for detailed analysis since interesting conditions triggered the event as explained
below. The COSMO model shows a good performance in simulating precipitation, hail and lightning over the Alpine region.
However, at the same time, it underestimates precipitation and lightning over the Adriatic Sea. As for the previous case, we
first look at the structure and evolution of the pre-storm environments using radiosonde profiles, but this time at the Milano
station (since it is closer to the event) in the north-western section of the Po Valley in Italy (Fig. 10a-b). At 12 UTC on 24
June 2017, a typical "loaded gun" structure can be identified with a temperature inversion at 850 hPa and dry air located above
warm and moist air and below an elevated mixed layer. Such a profile, also called "capping" layer is described in Lanicci
and Warner (1991), and is an indication of a severe storm environment. It is known as a type 1 tornado sounding. The cap or
lid prevents deep vertical mixing and inhibits the premature release of the convective instability with observed CIN of -144 J
$\mathrm{kg}^{-1}$ and CAPE as high as $2723 \, \mathrm{J} \, \mathrm{kg}^{-1}$. The CIN gradually decreased in the following hours without much change in CAPE
(not shown). At 00 UTC on 25 June 2017, the air becomes warmer below the lid, which indicates the deepening of warm and
moist air capped by a dry layer. This very unstable condition was favorable for deep convective development, and the potential
instability ahead of an upper-level trough lifted the air and released the accumulated energy. This led to a burst of thunderstorms
that hit this area in the early morning and produced a record-breaking amount of precipitation.

A comparison of the simulated profile at the Milano station (Fig. 10b) with the observed and above-discussed profile
(Fig. 10a) reveals a good performance of the model in capturing the vertical profile and thus the triggering mechanisms of
the event. The model reproduced the "capping" layer on the day before the event occurred and the deepening of the moist and
warm air several hours before the event occurred.

Analysis of the model output shows a warm and moist layer over the Po Valley with simulated total water content larger than
$45 \, \mathrm{kg} \, \mathrm{m}^{-2}$ (Fig. 10c-d). Given the southeasterly flow, a line of organized convection gradually formed along the northwest-
southeast oriented mountain edge at around 02:30 UTC (not shown). Subsequently, the convective cells continuously developed
over the elevated terrain and propagated upwind. Heavy precipitation was localized and became most intense between 03 and

**Figure 9.** Detailed characteristics of the case 23 July 2009. Thermodynamic skew-T log-P diagrams of (a) sounding observations and (b) COSMO extracted profiles at Payerne station at 00 (solid) and 12 UTC (dashed). Red and green lines represent the temperature and dew-point temperature profiles, respectively. Corresponding wind hodographs, shown in the bottom left corner, are obtained for 12 UTC on 23 July 2009. (c) COSMO simulated total water content and vertically integrated water flux vectors, (d) 0-6 km bulk wind shear, (e) temperature at 700 hPa, and (f) simulated reflectivity and wind barbs at 1 km above ground level at 12 UTC on 23 July 2009. The red box A1 indicates the zoomed subdomain shown in (f).

04 UTC with associated hail and lightning (Fig. 10e). This back-building process (e.g. Lagasio et al. (2017)) is shown with cross-sections normal to the squall line (Fig. 10f-h), where the convection developed upwind over the foothills. The triggered cells remain nearly stationary and the intensity significantly weakened when it moved to the northeast due to the loss of low-level warm and humid air over the Po Valley.

**Figure 10.** Similar as Fig. 9, but for the case of 25 June 2017. (a) Sounding observations and (b) COSMO extracted profiles at Milano station at 12 UTC on 24 June 2017 (solid lines) and 00 UTC on 25 June 2017 (dashed lines). (c) COSMO simulated total water content and vertically integrated water flux vectors, (d) 2 m temperature at 03 UTC on 25 June 2017, and (e) footprints of LPI and HAILCAST obtained between 03 and 04 UTC are shaded in yellow and purple, respectively. The red box B3 in (c) indicates the zoomed subdomain shown in (d-e). (f)-(h) Vertical cross-sections of potential temperature (gray contours), equivalent potential temperature (red contours), specific humidity (blue shaded), and simulated reflectivity (colour shaded) along the red transect B1-B2 at 02, 04 and 06 UTC on 25 June 2017.



### 3.4.3   The case of 8 July 2017 – thunderstorms near the Jura mountain

The case of 8 July 2017 is characterized by multiple thunderstorms over the Alps. Overall, the precipitation structure for this case is well reproduced, while the intensity is slightly underestimated with a large location error, which is most likely due to the southerly shift or the underestimation of the precipitation system. We again start with a look into the thermodynamic environment with the help of sounding observations at Payerne (Switzerland) near the location of hail occurrence (Fig. 11a-b). At 00 UTC, the profile shows a dry layer below 950 hPa capped by a moist layer or most probably by a cloud at around 800 hPa, which is topped by a dry layer above of 650 hPa (Fig. 11a). The observed CIN amounted to -130 J kg$^{-1}$ and CAPE to only 70 J kg$^{-1}$, which is not a favourable environment for thunderstorm development. In the morning hours, the stable layer was eroded away due to the strong upper-level westerly flow (see Fig. 2), making the conditions more favourable for the development of convection. At 12 UTC, a deep and well-mixed boundary layer was observed up to 800 hPa, nearly following the dry adiabatic profile. Comparison with the model (Fig. 11b) reveals that the model captures the vertical profile, even though temperature and dew-point temperature do not come as close as in observations at around 800 hPa level.

Fig. 11d shows a band of very low relative humidity at the 500 hPa level, consistent with a stratospheric intrusion embedded in the strong upper-level westerly flow (Fig. 2e). This band is near the Stuttgart sounding, but slightly to the south of it. There is also a pronounced upper-level cut-off low over Spain. Further analysis based on the model output, reveals that the westerly flow affected the northern pre-Alpine region (Fig. 11c), eroded the stable layer near the surface, and brought moisture to the northern Alpine foreland. Vertical cross-sections across this area at 05 UTC show that this dry and cold air was superimposed above the warm and moist layer near the surface (Fig. 11g). With significant instability, convection was initiated when the cold upper-level was advected over the warm near-surface air (Fig. 11h). Due to the dry and cold air aloft, evaporative cooling and melting of hydrometeors resulted in stronger and colder downdrafts. Several isolated cold pools that spread radially away with temperature depressions of 4 K can be identified from the subdomain C3 at 925 hPa level (Fig. 11e). The cold pool induced a large updraft velocity at the boundaries, which is favorable for convective intensification and new cell formation (Fig. 11f). Previous studies showed that the modeling framework is able to capture such developments (Leutwyler et al., 2016). On the other side, several convective cells formed in the late afternoon over the mountains and gradually formed two supercells (Fig. 11i). Several wet downbursts were confirmed by reliable sources (http://www.sturmarchiv.ch/index.php/Hagel) in central Switzerland (canton of Bern over Roggwil (around 14 UTC) and Wilderswil (around 15 UTC)) with wind gusts above 90 km h$^{-1}$.

## 4   Conclusions

In this study, we analyzed the simulations of eight observed cases of severe convection. The simulations were performed using a regional climate model, COSMO, at 2.2 km horizontal grid spacing, integrated with HAILCAST and LPI diagnostics over the Alpine-Adriatic region. The performance of the model in simulating precipitation, hail and lightning was evaluated against available observations. The main findings are summarized as follows.



**Figure 11.** As in Fig. 9, but for the case of 8 July 2017. (a) Sounding observations and (b) COSMO extracted profiles at Stuttgart at 00 (solid lines) and 12 UTC (dashed lines). (c) COSMO simulated total water content, (d) relative humidity at 500 hPa, (e) temperature at 925 hPa, and (f) vertical velocity at 850 hPa at 12 UTC on 8 July 2017. The red box C3 in (c) indicates the zoomed subdomain shown in (e), and the box C4 in (d) indicates the zoomed subdomain shown in (f). (g)-(i) Vertical cross-sections of humidity, temperature (red isolines) and simulated radar reflectivity along the red transect C1-C2 in (d).



Overall, the COSMO model together with HAILCAST and LPI diagnostics performed well in simulating precipitation, hail and lightning. In particular, the case-study simulations captured the main characteristics of the cases considered, such as the large-scale precipitation distributions, or the occurrence of elongated hail swaths versus localized hail events controlled by
topography (Fig. 4-7). The best performance was obtained for the cases with strong synoptic forcing. This is to some extent associated with the chaotic nature of the underlying dynamics and the lower predictability of these kinds of events. The two events associated with the strongest synoptic forcing (1 June 2013 and 17 May 2018) are events with heavy precipitation (especially 1 June 2013), but with no or very little hail and lightning. Even though the model overestimated the precipitation intensity for these two events, it produced no or very little hail, which is in accordance with the observations. Overall, we see
that the performance in the simulation of hail and lightning is consistent with the model performance for precipitation. Comparison of the model with radar-based hail estimates revealed that COSMO integrated with HAILCAST tends to underestimate the frequency of large hailstones, and fails to produce extra-large hailstones (larger than 40 mm). However, when compared to crowd-sourced and hailpad observations, it shows a good distribution. It is possible that some of the biases could be addressed by tuning the diagnostic computations of hail and lightning.

The ability of COSMO to simulate severe convective events associated with hail and lightning enables further exploration of the mechanisms that drive such events. By investigating three specific cases that are selected according to their impacts over different severe weather hot spots of the Alpine-Adriatic region, we identified several storm environments that contribute to heavy precipitation, hail and lightning. These mechanisms include a capping layer that serves to accumulate humidity and energy below this layer (23 July 2009, 25 June 2017), a "back building process" that contributes to convective cells that remain
quasi-stationary near elevated terrains (25 June 2017), dry air above a warm and moist surface that leads to higher instability and stronger downdrafts (8 July 2017), and an upper-level trough that promotes ascent (25 June 2017). The results show that, although the simulations are not designed to simulate the detailed structure, amplitude and location of the events in terms of precipitation, hail and lightning, COSMO is generally able to credibly replicate key processes of severe thunderstorms and create the related favourable environments for storm development.

Our findings show that HAILCAST and LPI integrated with COSMO are promising tools to diagnose hail and lightning over the Alpine Adriatic region (as also shown by (Malečić et al., 2023)). However, a couple of shortcomings are revealed: (i) Comparison of the model to available hail observations reveals that COSMO HAILCAST fails to reproduce extra-large hailstones. The most likely cause for the lack of large hailstones is the underestimation of strong updrafts in COSMO. Such an underestimation is plausible, as with a computational resolution of 2 km, simulations of heavy convection exhibit signs of bulk
converge, but not yet structural convergence (Panosetti et al., 2018). In other words, the horizontal scales of the thunderstorm are overestimated, and peak updrafts are underestimated. In this context, wind shear is important as it influences the convective dynamics and the generation of rotating thunderstorms with strong updrafts. (ii) The spatial extent of hail footprints are overestimated in COSMO HAILCAST compared to the radar-based observations. This could be due to the fact that MESHS only provides the estimation of hailstones larger than 20 mm, while POH only provides the probability of hail. (iii) The output
of HAILCAST is sensitive to the initial hail embryo (e.g., the maximum hail diameter always comes from the largest hail embryo) as shown by Adams-Selin and Ziegler (2016); Adams-Selin et al. (2019). (iv) For the LPI, the threshold of vertical



velocity is resolution dependent (Brisson et al., 2021), thus a comprehensive analysis against observations is required before application. The LPI provides the potential of lightning, not the exact number of lightning flashes, which makes it difficult to evaluate against observations. Thus our analysis was only focused on the coverage or footprints of the lightning.

The promising results of these case studies reveal that the model is not only able to simulate critical impacts of severe convective events, but is remarkably able to also simulate their drivers. This gives us confidence in future applications of hail and lightning diagnostics for climate simulations and analyses of potential changes with further warming of the atmosphere, which is part of a parallel ongoing study.

*Author contributions.* RC performed the simulations and analysed the model output and observations. RC, NB, MED and CS wrote the
manuscript. NB, MED and CS provided scientific advice throughout the project.

*Competing interests.* The authors declare that they have no conflict of interest.

*Acknowledgements.* The authors acknowledge the Partnership for advanced computing in Europe (PRACE) for awarding us access to Piz Daint at ETH Zürich/Swiss National Supercomputing Centre (CSCS, Switzerland). We also acknowledge the Federal Office for Meteorology and Climatology (MeteoSwiss), CSCS, the Center for Climate Systems Modeling (C2SM) and ETH Zürich for their contributions to
the development and maintenance of the GPU-accelerated version of COSMO. Lightning data is obtained from the Lightning Detection Network in Europe (LINET, https://www.nowcast.de/en/solutions/linet-data). The authors would also like to acknowledge MeteoSwiss for providing radar (POH, MESHS) and rain gauge (RhiresD) observations, NASA for providing the IMERG data, University of Wyoming for providing sounding observations, the Mobiliar Lab for Natural Risks of the University of Bern for providing the crowd-sourced hail reports, and members from the department of Geophysics of the University of Zagreb for providing the hailpad observations. We acknowledge
Copernicus Climate Change Service (C3S) for generating ERA5 dataset, and the German Climate Computing Center (DKRZ) for providing post-processed COSMO-ready ERA5 reanalysis boundary conditions. RC thanks Barbara Malecić and Damjan Jelić for fruitful scientific discussions that contributed to the analyses of this evaluation.

*Financial support.* The work presented here has been conducted within the SWALDRIC (180587) project of the Croatian-Swiss Research Programme (CSRP) funded by the Swiss National Science Foundation (SNSF) and the Croatian Science Foundation (CSF).



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
