# Peer review of "Exploring hail and lightning diagnostics over the Alpine-Adriatic region in a km-scale climate model"

_Weather and Climate Dynamics, 2023_

## Referee Comment (RC2)

Review of: "Exploring hail and lightning diagnostics over the Alpine-Adriatic region in a km-scale climate model" by Cui et al.

**Recommendation:** Minor revisions

**Summary**

The authors compare 2.2 km COSMO simulations driven by reanalysis data with different observations of total precipitation, hail, and lightning of a number of severe weather events. Overall, the paper is written sufficiently clearly, and I only have a (relatively large number of) somewhat minor comments.

**Main comments**

1. It is not quite clear what the goal of the paper is: Assess the COSMO diagnostics or to explain why severe convective weather occurred on those days. I have to admit that I didn't get much out of the short case studies (which are partially redundant with the case summaries earlier in the paper); is there anything new to be learned from these?

2. Throughout the manuscript, there are statements touting the simulations as performing "very well." I think such statements are subjective and should be avoided unless a metric has been defined and justified, which quantifies the performance of the simulations. I suggest deleting these statements and simply report the differences between the observations and simulations, and let the reader decide to what extent the model diagnostics are useful (or maybe offer your take on the results in the conclusions section without overselling the quality of the simulations).

**Specific comments**

1. Line 4, and throughout: Since you are still in the km scale, the simulations are *convection allowing*, but not *convection resolving* (to resolve convective flows, a grid spacing of O(100) m is required).

2. Line 8: Write out the COSMO acronym before using it the first time.

3. Line 13: Is the topographic barrier a requirement, or is it just present? This feature does not seem to be highlighted later on, and I'm not sure it has been demonstrated that this barrier was indeed a cause of the severe weather. More analysis/discussion would be needed to grant this feature a place in the abstract.

4. Line 25: Reword thunderstorm peril region (e.g., regions at highest risk of experiencing thunderstorms).

5. Line 44: Can you be more specific about which aspects are not well-understood? Such broad statements tend to dismiss the wealth of research that has been done in this area.

6. Line 47: Wind shear also promotes the organization of quasi-linear systems and updraft intensification (e.g., Markowski and Richardson 2010).

7. Line 49: One would expect a given environment to lead to the same storm, whether or not the storm is surrounded by mountains. Can you specify what you mean by stating that the role of the ingredients is complicated? Do you mean that there is a larger suspected variability of the thermodynamics/kinematic fields compared to flat terrain, or that the environment cannot easily be sampled (because it may be rather inhomogeneous)? Or do you specifically talk about the lift ingredient (the subsequent text seems to imply such)?

8. Line 53: Grammar broken (…and subsequently to…)

9. Line 54: "The" mesoscale boundary hasn't been introduced yet (could say "a" mesoscale convergence zone).

10. Line 102: Isn't 1 h is the interval (rather than a frequency)?

11. Line 103: Add AGL/MSL after 23.5 km?

12. Line 114: Calculated instead of accumulated?

13. Line 123: Are there also non-convective thunderstorms?

14. Line 140: Maybe add that these values are much lower than what is observed in the real world because of the relatively coarse grid spacing (the convective clouds tend to be wider, and updrafts weaker than in truly convection resolving simulations).

15. Line 141: Why is CAPE negative?

16. Line 147: Delete spatial

17. Line 157: Please add the size in cm for each category

18. Line 164: Hailpad observations

19. Line 180: LINET data are not gridded (right?), so they can't really have a resolution. Perhaps say that the location error is 3 km? Why is the time known only to within 2 min? Given that the system has something like nano-second accuracy internally, 2 min seems like a huge inaccuracy.

20. Line 192: Write out the acronym before using the abbreviated form.

21. Line 198: Reword: e.g., …indicates that the model field is too spread-out/broad/diffuse. Also recommend avoiding these parenthetical constructions to shorten the sentence.

22. Line 225: "This cold front" hasn't been introduced yet

23. L. 229: Please specify what was discharged (water?)

24. Line 234: damage (instead of damages)

25. Line 265, 274: Model showing good performance: This seems like an entirely subjective statement. Either define what you consider as "good" or leave that judgement to the reader (and just report the errors).

26. Line 267: Reword: …captured relatively well

27. Line 288: Model reproducing hail "very well": Maybe the general presence of hail is predicted well, but the placement and coverage is (not surprisingly) not captured very well (e.g., 8 July 2017). Like above, I suggest reporting the differences and omit statements about the quality of the simulations.

28. Line 301: … hailstone **diameters** above 20 mm.

29. Line 315: Another instance of "good" model performance.

30. Line 321: Suggest replacing precipitation with rain or "total precipitation" (hail is also "precipitation").

31. Line 325: It is not surprising that the model is unable to produce accurate hail sizes; the updrafts are barely resolved, you are using single-moment microphysics, etc. I would probably consider omitting the hail size comparison, or at least state upfront that the model cannot be expected to produce accurate hail sizes (forecasting hail size accurately remains a holy grail of severe weather prediction).

32. Line 375: If the situation is strongly synoptically-forced, then presumably there is a good amount of flow in the troposphere, typically leading to fast storm motions. Why do the storms become quasi-stationary?

33. Line 444: True, but updrafts also suffer more from entrainment, so the net effect tends to be less evaporative cooling (James and Markowski, 2010, MWR).

**Figures**

Fig. 4: Consider adding labels to the rows to make it easier to identify which panels refer to the observations, and which to the simulations

Fig. 5: The panels, and especially the legends are too small; consider breaking this figure into two.

---

## Author Comment (AC1)

**Response to Reviewer 1**

**Review of manuscript WCD-2023-11 "Evaluation of hail and lightning climatology using km-scale climate model over the Alpine region"**

by Ruoyi Cui, Nikolina Ban, Marie-Estelle Demory, Raffael Aellig, Oliver Fuhrer, Jonas Jucker, Xavier Lapillonne, Christoph Schär

The study presents simulations of severe thunderstorms in the Alpine and Adriatic areas. The eight case studies are run with ERA-5 boundary conditions and initiated at 12 UTC of the previous day to allow for spin-up. The model simulations include, for the first time HAILCAST and LPI diagnostics to identify and characterize hail and lightning over the study areas in Europe. The simulations are compared in detail to observations.

The paper is well-written, it is very well structured, and the figures are very clear and well-readable. The results are highly relevant, and the detailed process discussions are instructive and illustrate the diversity of thunderstorm environments in the complex topography. The paper is ready for publication after some minor revisions.

We thank Olivia Martius for her valuable comments and positive feedback on our study. Below we list the replies to her comments in blue.

**Major points**

L312: I think that there is an issue with this comparison. If you compare two relative frequency distributions that both sum up to 1 for a variable that is expected to have an exponential distribution and one distribution starts at 0 mm and one at 20 mm, then by construction you will always find relatively fewer large values in the distribution that starts at 0 mm. To check if this is true, you can artificially cut the model data at 20 mm. Since the issue is not present in the comparison with the crowd-sourced reports, I suspect that your comparison improves a lot if you use only HAILCAST data of >= 20 mm.

Thank you for pointing this out. The reviewer is correct. As the two distributions cover a different range in hail diameter, the normalization used in the submitted version of the paper leads to artifacts. The reviewer suggested restricting the analysis to the range with hail diameters > 20 mm (as available in both MESHS and HAILCAST). In the revisions, we choose an alternative approach, i.e. considering the area affected by hail rather than the frequency. This resolves the issue and makes the two displayed distributions physically comparable. The revised results are shown in Figure R1 below. With these revisions, the two distributions are rather similar over the range covered.

With this approach, we don't need to consider the issue of different resolutions between the model output and observational data (e.g. area = number of grid points falling within each bin×area of obs/mod grid point). The area where the model is able to simulate hailstone below 20 mm is also shown in this figure. We plan to revise the text accordingly throughout the manuscript: HAILCAST underestimates the area affected by large hailstones compared to MESHS observations. As the panels (a-h) present different comparisons (e.g., area and frequency), we plan to break this figure into two figures.

**Minor points**

L20: e.g. Punge and Kunz...

Added.

L30: averaged –> average

[Figure]

**Figure R1:** Daily area affected by hail for each size bin obtained in MESHS observations (orange) and COSMO simulations (blue) for all cases over the radar domain.

Changed.

L31: is located between

Changed.

L33: A transition sentence is needed before the sentence "It remains..."

The sentence has been modified in the following way: "However, owing to difficulties in observing and modeling such events, it remains a challenge to understand the characteristics and mechanisms of such severe convective events."

L35: local-scale

Changed.

L39: With this –> Radar provides

Changed.

L45: due to multiple

Changed.

L49: The statement "has been used . . . " is very vague, maybe be a bit more specific.

Changed the sentence to "[...], has been used to identify atmospheric conditions prone to severe convective storms."

L59: troughs

Changed

L119: Please specify in more detail the variable that hailcast stores at hourly intervals. Is it the mean, max or instantaneous hailstone size?

Changed the last sentence in this paragraph to "The hourly maximum hail diameter among the five prescribed hail embryos is stored at hourly output intervals, providing information on hail swaths and the maximum expected hail size over an hour."

L131: is epsilon a function of height?

No, $\epsilon$ is a dimensionless number that has a value between 0 and 1 defined in Equation 2. It scales the cloud updrafts and reaches the maximum when the vertically averaged mixing ratios of $q_L$ and $q_F$ are equal. We changed the sentence on L131 in the revised manuscript.

L150: Maybe add a sentence on the ability of RhighresD to capture convective precipitation? Local maxima could be missed due to too coarse station density.

Thank you for pointing this out. We added the discussion of RhiresD quality in the following way on L153: "The dataset suffers from a general tendency to overestimate light precipitation and underestimate intense precipitation due to interpolation uncertainty. The uncertainty is higher in data-sparse areas and in cases of high spatial variation (e.g. convective rainfall)."

L153: in-situ observations and remote radar-based observations

Changed.

L167: crowd-sourced reports –> insurance loss reports

Changed.

L169: delete "on the other side" since there is no matching on the one side.

Deleted.

L169: in units of cm starting at 20mm :-)

Changed the unit of hail diameter to mm throughout the manuscript. We corrected this sentence to "MESHS estimates the largest expected hail diameter starting at 20 mm."

L172: 2009 –> 2002

Changed to "[...] C-band radars (in operation since 2002) and [...] (Nisi et al., 2016)"

L186: we use data from

Changed.

L200ff: Please explain the calculation of tau in a bit more detail. Is dCAPE/dt a daily mean or daily max value?

The calculation of $\tau$ is described in the L210. We would consider move this part a couple of lines upward from the previous location to provide the information a bit earlier.

L223: heavy –> severe

Changed.

L255: near the mountainous region of central Europe –> a bit more specific?

Changed to "eastern France and the southern flank of the Alps".

L265: Please explain in a bit more detail how you compare across the different spatial resolutions and how sensitive the method is to the interpolation in space. I suggest adding 1 – 2 sentences of discussion.

For a fair comparison, both observations and simulations are interpolated to the same grid with 12 km grid spacing. This is described in Section 2.5. Please see L191.

L298: Note that the black forest is already quite far away from the radar, and the data quality may become an issue because the radar gates become very big and the lowest beam height is quite high. If you compare only within a 140 km area around the radars, the radar quality should be very good.

We appreciate your input. Indeed, the data quality might be an issue due to beam-widening effects. While there could be a trade-off between the limitation and breadth of the analysis, we believe that the

benefits of a broader range could outweigh the potential marginal if we limit the radius within a 140 km radius. Therefore, we suggest maintaining the current figure to allow us to glean insights into model performance to a broader extent.

L303: Please mention that both radar-based hail products only see hail in the air and not on the ground.

Added the sentence on L305: "Furthermore, both MESHS and POH detect hail at higher altitudes and not on the ground, and melting may influence small hailstones (Nisi et al., 2016)."

L371: Please mention the time of the hailfall at the beginning of the case study description.

Thank you for your suggestion. We plan to add the time of hail fall from observations in the revised manuscript.

L380ff: I do not yet fully get my head around the timing of the individual steps and the position of the cold front. You mention the advection of cold air; this would happen behind the cold front. But then you mention the increased dew point ahead of the cold front. It would be great to know where the front is, at which altitude, and at which time to get the full (complex) 4-D picture of the event.

We agree that there might be some confusion with the description. Please find Figure R2 below which shows the 500 hPa geopotential height and 850 hPa temperature for the case 23 July, 2009, from ERA5 reanalysis. Regarding the dew point temperature question, we changed the sentence on L378: "Fig. 9a-b shows the observed and modeled Skew-T log-p thermodynamic diagram from Payerne at 00 and 12 UTC on 23 July 2009, which provides information where Switzerland was ahead and behind the cold front". And accordingly, change the sentence on L384: "[...] behind the cold front". We plan to add the location of the front, altitude and time in the revised manuscript.

[Figure]

**Figure R2:** Synoptic situation for 23 July 2009. Panels show geopotential height at 500 hPa (m, black contours), temperature at 850 hPa (°C, shaded) obtained from ERA5 reanalysis.

L409: becomes warmer –> I do not see this; to me, the temperature near the surface looks colder at 00 UTC than at 12 UTC.

Thank you for pointing this out. We changed the sentence to: "At 00 UTC on 25 June 2018, the air below 900 hPa became cooler and the lid was higher, which indicates the capped inversion was broken. Together with the potential instability ahead of an upper-level trough, the buoyant air was lifted and released the accumulated energy. This led to a burst of thunderstorms that hit this area in the early morning and produced a record-breaking amount of precipitation."

L418: Given the southeasterly flow –> this implies causality; please explain in more detail.

Changed the text accordingly: "As shown in Fig. 2b, due to the presence of an upper-level trough, the Po Valley was influenced by the southwesterly flow. A line of [...]".

L432: Please explain in a bit more detail why the stable was eroded by the upper-level westerly flow.

Changed the text accordingly: "The stable layer was eroded away due to the warming of the near-surface air in the morning hours, making the conditions [...]".

L439: add not shown

Are you referring to the cut-off low over Spain? This can be referred to Fig. 2e.

L461: It is unclear to what "this" refers and what is meant by these kind of events (I think there is some grammatical confusion as to whether you mean the synoptically driven cases or not)

Changed the sentence to: "This is to some extent associated with the chaotic nature of the underlying dynamics and the lower predictability of the localized events."

L471: three cases that were selected

Changed

L485: In this context... this statement is not very clear to me yet

We plan to delete this sentence.

**Figures**

Fig.1: caption four –> three Changed.

Fig.11: text says the sounding is at Payerne, but the figure title suggests the sounding is at Stuttgart.

We apologize for this confusion, and the text should be Stuttgart. We can see the stratospheric intrusion better at Stuttgart compared to Payerne, because it's closer to the hailstorm. The text will be corrected accordingly.

Fig.11: Please add the borders of Switzerland.

We have the borders of Switzerland in Fig. 9-11. Do you mean to make the borders thicker?

**References**

Nisi, L., Martius, O., Hering, A., Kunz, M., and Germann, U. (2016). Spatial and temporal distribution of hailstorms in the Alpine region: A long-term, high resolution, radar-based analysis. *Quarterly Journal of the Royal Meteorological Society*, 142(697):1590–1604.

**Response to Reviewer 2**

**Review of manuscript WCD-2023-11 "Evaluation of hail and lightning climatology using km-scale climate model over the Alpine region"**

by Ruoyi Cui, Nikolina Ban, Marie-Estelle Demory, Raffael Aellig, Oliver Fuhrer, Jonas Jucker, Xavier Lapillonne, Christoph Schär

The authors compare 2.2 km COSMO simulations driven by reanalysis data with different observations of total precipitation, hail, and lightning of a number of severe weather events. Overall, the paper is written sufficiently clearly, and I only have a (relatively large number of) somewhat minor comments.

We thank anonymous Reviewer 2 for his/her valuable comments and positive feedback on our study. Below are the replies to his/her comments in blue.

**Major points**

It is not quite clear what the goal of the paper is: Assess the COSMO diagnostics or to explain why severe convective weather occurred on those days. I have to admit that I didn't get much out of the short case studies (which are partially redundant with the case summaries earlier in the paper); is there anything new to be learned from these?

Thank you for your comments. We agree that the short case studies might not fully address the second objective on L84. We try to update the three listed objectives to two: (1) Evaluate the performance of the COSMO model at km-scale grid spacing in simulating hail and lightning; (2) Explore how storm environments are represented by the COSMO model through case studies.

Throughout the manuscript, there are statements touting the simulations as performing "very well." I think such statements are subjective and should be avoided unless a metric has been defined and justified, which quantifies the performance of the simulations. I suggest deleting these statements and simply report the differences between the observations and simulations, and let the reader decide to what extent the model diagnostics are useful (or maybe offer your take on the results in the conclusions section without overselling the quality of the simulations).

Thank you for your feedback. We appreciate your perspective regarding the statements that may appear subjective, illustrating the simulations as performing "very well". To address this concern, we revised the manuscript by removing the subjective statements and focusing on reporting the difference between observations and simulations. We have reviewed the examples provided in the minor points section and addressed each one individually, please have a look.

**Minor points**

L4ff: Since you are still in the km scale, the simulations are convection allowing, but not convection resolving (to resolve convective flows, a grid spacing of O(100) m is required).

Changed the sentence to: "These kilometer-scale models improve [...]". Later, use km-scale models throughout the manuscript.

L8: Write out the COSMO acronym before using it the first time.

Changed the sentence to: "The simulations are performed with the climate version of regional model Consortium for Small-scale Modeling (COSMO) that runs on Graphics Processing Units (GPUs) at a horizontal grid spacing of 2.2 km."

L13: Is the topographic barrier a requirement, or is it just present? This feature does not seem to be highlighted later on, and I'm not sure it has been demonstrated that this barrier was indeed a cause of the severe weather. More analysis/discussion would be needed to grant this feature a place in the abstract.

The topographic barrier mentioned here does play a crucial role in shaping the local atmospheric conditions for severe weather, for example, the 25 June 2017 morning case over the Po Valley. We added a sentence on L424 to highlight this topographic feature: "The topography of the Po Valley offers a favorable environment for the initiation of new cells, which consequently explains the occurrence of the hot spot."

L25: Reword thunderstorm peril region (e.g., regions at highest risk of experiencing thunderstorms).

Changed the sentence to: "It is recognized as one of the regions at high risk of experiencing thunderstorms in Europe due to its notable topography."

L44: Can you be more specific about which aspects are not well-understood? Such broad statements tend to dismiss the wealth of research that has been done in this area.

Changed the first sentence to be more specific: "Although severe convective storms can cause catastrophic damage, important processes for the prediction of hail and lightning are insufficiently represented in weather and climate models."

L47: Wind shear also promotes the organization of quasi-linear systems and updraft intensification (e.g., Markowski and Richardson 2010).

Added the sentence and reference to L47 as suggested.

L49: One would expect a given environment to lead to the same storm, whether or not the storm is surrounded by mountains. Can you specify what you mean by stating that the role of the ingredients is complicated? Do you mean that there is a larger suspected variability of the thermodynamics/kinematic fields compared to flat terrain, or that the environment cannot easily be sampled (because it may be rather inhomogeneous)? Or do you specifically talk about the lift ingredient (the subsequent text seems to imply such)?

We agree the statement is quite vague. We were trying to list the ingredients for hailstorms and which of them are used as proxies to diagnose hail. L49 is now expanded to clarify that: Furthermore, over complex topography, additional mechanisms may also affect the initiation and development of convection. For example, as terrain increases, deep layer vertical level shear increases and CAPE decreases, and cold pools are blocked and become stronger and deeper (Mulholland et al., 2019).

L53: Grammar broken (. . . and subsequently to. . . )

Changed the sentence to: "[...], and subsequently to transport via upvalley winds to the mountain crest."

L54: "The" mesoscale boundary hasn't been introduced yet (could say "a" mesoscale convergence zone).

Changed the mesoscale convergence zone to a mesoscale convergence zone.

L102: Isn't 1 h is the interval (rather than a frequency)?

Changed updating frequency to updating interval of 1 hour.

L103: Add AGL/MSL after 23.5 km

Changed the sentence to: "[...], where vertical spacing ranges from 20 m near the surface to 1.2 km at the model top located around 23.5 km above mean sea level."

L114: Calculated instead of accumulated?

Updraft duration is calculated, we changed the explanation to make it clearer: If the grid column or any adjacent grid columns has a maximum updraft exceeding $10\,\mathrm{ms}^{-1}$ between the previous and current model time steps, the updraft duration field is incremented by one model time step. It is used to track the convective cells and limit the maximum updraft time in the hail model.

L123: Are there also non-convective thunderstorms?

Actually, there are also non-convective thunderstorms (e.g. spider lightning, dry lighting, etc.), mostly cloud-to-ground lightning. As this statement was taken from Lynn and Yair (2010) and LPI is usd for convective thunderstorms (via the non-inductive mechanism), we have decided to keep the sentence as it is. The case studies from Yair et al. (2010) illustrates that WRF-calculated LPI might not predict lighting caused by stratiform precipitation very well.

L140: Maybe add that these values are much lower than what is observed in the real world because of the relatively coarse grid spacing (the convective clouds tend to be wider, and updrafts weaker than in truly convection resolving simulations).

Thanks for pointing it out. We added the sentence as you suggested on L141.

L141: Why is CAPE negative?

The value -1500 J $\mathrm{kg}^{-2}$ here is the threshold for the filter $f_2$ discussed in the appendix B (Equation 16) in Brisson et al. (2021). The equation is formally similar to mixed-layer CAPE but with a different integration layer. It is used to remove the spurious LPI signals under the conditions of gravity waves embedded in moist flow (e.g. during Föhn events in the Alps in winter). It is approximately 0 or slightly negative at locations of explicitly simulated convective cells, but attains large negative values for the stable conditions associated with orography mountain waves. We added: "see Eq. 16 in Brisson et al. (2021)" on L142 to provide detailed information for this integrated buoyancy.

L147: Delete spatial Changed.

L157: Please add the size in cm for each category.

Added the size in mm accordingly on L159-164, and decided to change the units to mm throughout the manuscript to be consistent.

L164: Hailpad observations. Changed.

L180: LINET data are not gridded (right?), so they can't really have a resolution. Perhaps say that the location error is 3 km? Why is the time known only to within 2 min? Given that the system has something like nano-second accuracy internally, 2 min seems like a huge inaccuracy.

LINET are not gridded, in fact, it has an average location accuracy of approximately 150 m. The LINET data used in this study is taken from Jelić et al. (2020). It was regridded to 3 km grid spacing with a temporal resolution of 2 min. Higher temporal and spatial resolutions are possible, while the database exceeds the computational and storage resources. The regridded 2D database still provides sufficient information to discern the local characteristics. Later, we aggregated the lightning flashes every hour in order to compare against simulated hourly maximum LPI.

L192: Write out the acronym before using the abbreviated form.

Changed the sentence to: "We then use the SAL (Structure-Amplitude-Location) method [...]."

L198: Reword: e.g., ...indicates that the model field is too spread-out/broad/diffuse. Also recommend avoiding these parenthetical constructions to shorten the sentence.

Changed the sentence to: "The A component is calculated as the normalized difference between the domain average from observed and simulated fields, with positive (negative) values indicating overestimation (underestimation) by the model. The L component takes into account the displacement of the center of mass between the observed and simulated field, as well as the weighted-average

distance between individual objects and the mass center of the total field. A lower L value indicates a good representation of the model field. The S component considers the size and shape of the objects, positive values suggest a more widespread, while negative values indicate a more peaked field."

L225: "This cold front" hasn't been introduced yet

Thank you for your comment. As also suggested by review 1, we have included a description of the front's location.

L229: Please specify what was discharged (water?)

Changed the text to: "[...] caused water discharges with return periods of 10 to 30 years [...]."

L234: damage (instead of damages) Changed.

L265, 274: Model showing good performance: This seems like an entirely subjective statement. Either define what you consider as "good" or leave that judgment to the reader (and just report the errors).

We tried to rewrite the text to illustrate the performance instead of making subjective statements:
L265: As shown in the SAL diagram of daily accumulated precipitation (Fig. 3), the amplitude [...].
L274: Figure 4 depicts the patterns of daily accumulated precipitation in comparison to the IMERG and RhiresD observations. The best performance among eight cases is seen [...].

L267: Reword: ...captured relatively well

Changed.

L288: Model reproducing hail "very well": Maybe the general presence of hail is predicted well, but the placement and coverage is (not surprisingly) not captured very well (e.g., 8 July 2017). Like above, I suggest reporting the differences and omit statements about the quality of the simulations.

Changed the sentence on L288: "The general presence of hail is simulated well, but the placement and coverage is not surprisingly not captured very well (Fig. 5)."

L301: ... hailstone diameters above 20 mm.

Changed.

L315: Another instance of "good" model performance.

Changed the sentence to: "Moreover, compared to hailpad observations over Croatia (panels l-n), the model shows a distribution that aligns closely with the observed data, particularly for the two cases of 25 June 2017 and 17 May 2018. While the case of 24 July 2017 (an event relatively well captured by the model according to the previous results) shows underestimated simulated hail sizes."

L321: Suggest replacing precipitation with rain or "total precipitation" (hail is also "precipitation").

We plan to replace all the precipitation with total precipitation.

L325: It is not surprising that the model is unable to produce accurate hail sizes; the updrafts are barely resolved, you are using single-moment microphysics, etc. I would probably consider omitting the hail size comparison, or at least state upfront that the model cannot be expected to produce accurate hail sizes (forecasting hail size accurately remains a holy grail of severe weather prediction).

We agree that the model cannot be expected to produce accurate hail sizes. However, it could be useful to show the comparison against available observations. As raised by Reviewer 1, it's unfair to compare the frequency from HAILCAST and MESHS, because MESHS only provides an estimation of hail size >20 mm. Instead, we plan to change the panel (a-h) in Fig. 6 by comparing the area affected by hail from observations and the model. The updated panels (a-h) are shown in Figure R1.

L375: If the situation is strongly synoptically-forced, then presumably there is a good amount of flow in the troposphere, typically leading to fast storm motions. Why do the storms become quasi-stationary?

We agree that this could lead to confusion, and this sentence is changed to: "The hail occurrence is located over areas where fronts typically become quasi-stationary along the Alpine foothills, due to the distortion of the flow field (Schumann, 1987). The propagation of the front is slow, while the convergence along the front results in a fast storm movement."

L444: True, but updrafts also suffer more from entrainment, so the net effect tends to be less evaporative cooling (James and Markowski, 2010, MWR).

Thank you for this comment and the reference. We plan to reformulate this sentence and reconsider the impact of the dry air aloft on downdrafts and cold pools.

**Figures**

Fig.4: Consider adding labels to the rows to make it easier to identify which panels refer to the observations, and which to the simulations.

Thank you for your suggestions. We added labels to the rows to make it more reader-friendly.

Fig.5: The panels, and especially the legends are too small; consider breaking this figure into two.

Thank you for your suggestions. We plan to adjust the size of the legends.

**References**

Brisson, E., Blahak, U., Lucas-Picher, P., Purr, C., and Ahrens, B. (2021). Contrasting lightning projection using the lightning potential index adapted in a convection-permitting regional climate model. *Climate Dynamics*, 57(7-8):2037–2051.

Jelić, D., Megyeri, O. A., Malečić, B., Belušić Vozila, A., Strelec Mahović, N., and Telišman Prtenjak, M. (2020). Hail Climatology Along the Northeastern Adriatic. *Journal of Geophysical Research: Atmospheres*, 125(23).

Lynn, B. and Yair, Y. (2010). Prediction of lightning flash density with the WRF model. *Advances in Geosciences*, 23(8):11–16.

Mulholland, J. P., Nesbitt, S. W., and Trapp, R. J. (2019). A Case Study of Terrain Influences on Upscale Convective Growth of a Supercell. *Monthly Weather Review*.

Schumann, U. (1987). Influence of mesoscale orography on idealized cold fronts. *Journal of the Atmospheric Sciences*, 44(23):3423–3441.

Yair, Y., Lynn, B., Price, C., Kotroni, V., Lagouvardos, K., Morin, E., Mugnai, A., and del Carmen Llasat, M. (2010). Predicting the potential for lightning activity in mediterranean storms based on the weather research and forecasting (WRF) model dynamic and microphysical fields. *Journal of Geophysical Research*, 115(D4).

---

## Author Response (AR1)

**Response to Reviewer 1**

**Review of manuscript WCD-2023-11 "Evaluation of hail and lightning climatology using km-scale climate model over the Alpine region"**

by Ruoyi Cui, Nikolina Ban, Marie-Estelle Demory, Raffael Aellig, Oliver Fuhrer, Jonas Jucker, Xavier Lapillonne, Christoph Schär

The study presents simulations of severe thunderstorms in the Alpine and Adriatic areas. The eight case studies are run with ERA-5 boundary conditions and initiated at 12 UTC of the previous day to allow for spin-up. The model simulations include, for the first time HAILCAST and LPI diagnostics to identify and characterize hail and lightning over the study areas in Europe. The simulations are compared in detail to observations.

The paper is well-written, it is very well structured, and the figures are very clear and well-readable. The results are highly relevant, and the detailed process discussions are instructive and illustrate the diversity of thunderstorm environments in the complex topography. The paper is ready for publication after some minor revisions.

We thank Olivia Martius for her valuable comments and positive feedback on our study. Below we list the replies to her comments in blue.

**Major points**

L312: I think that there is an issue with this comparison. If you compare two relative frequency distributions that both sum up to 1 for a variable that is expected to have an exponential distribution and one distribution starts at 0 mm and one at 20 mm, then by construction you will always find relatively fewer large values in the distribution that starts at 0 mm. To check if this is true, you can artificially cut the model data at 20 mm. Since the issue is not present in the comparison with the crowd-sourced reports, I suspect that your comparison improves a lot if you use only HAILCAST data of >= 20 mm.

Thank you for pointing this out. The reviewer is correct. As the two distributions cover a different range in hail diameter, the normalization used in the submitted version of the paper leads to artifacts. The reviewer suggested restricting the analysis to the range with hail diameters > 20 mm (as available in both MESHS and HAILCAST). In the revisions, we chose an alternative approach, i.e. considering the area affected by hail rather than the frequency. This resolves the issue and makes the two displayed distributions physically comparable. The revised results are shown in Figure R1 below. With these revisions, the two distributions are similar over the range covered.

With this approach, we don't need to consider the issue of different resolutions between the model output and observational data (e.g. area = number of grid points falling within each bin×area of obs/mod grid point). The area where the model is able to simulate hailstone below 20 mm is also shown in this figure. We revised the text accordingly throughout the manuscript: "HAILCAST underestimates the area affected by large hailstones compared to MESHS observations". As the panels (a-h) present different comparisons (e.g., area and frequency), we broke this figure into two figures.

**Minor points**

L20: e.g. Punge and Kunz...

Added.

L30: averaged –> average

[Figure]

**Figure R1:** Daily area affected by hail for each size bin obtained in MESHS observations (orange) and COSMO simulations (blue) for all cases over the radar domain.

Changed.

L31: is located between

Changed.

L33: A transition sentence is needed before the sentence "It remains..."

The sentence has been modified in the following way on L33: "However, owing to difficulties in observing and modeling such events, it remains a challenge to understand the characteristics and mechanisms of such severe convective storms".

L35: local-scale

Changed.

L39: With this –> Radar provides

Changed.

L45: due to multiple

We try to be more specific, therefore we change the sentence to "[...], important processes, such as hail growth (Adams-Selin, 2023) and lightning processes (Fierro et al., 2013), for predicting hail and lightning are insufficiently represented in weather and climate models".

L49: The statement "has been used . . . " is very vague, maybe be a bit more specific.

Changed the sentence to "[...], has been used to identify atmospheric conditions prone to severe convective storms".

L59: troughs

Changed

L119: Please specify in more detail the variable that hailcast stores at hourly intervals. Is it the mean, max or instantaneous hailstone size?

Changed the last sentence in this paragraph to "The hourly maximum hail diameter among the five prescribed hail embryos is stored at hourly output intervals, providing information on hail swaths and the maximum expected hail size over an hour".

L131: is epsilon a function of height?

No, $\epsilon$ is a dimensionless number that has a value between 0 and 1 defined in Equation 2. It scales the cloud updrafts and reaches the maximum when the vertically averaged mixing ratios of $q_L$ and $q_F$ are equal. We changed the sentence on L133 in the revised manuscript.

L150: Maybe add a sentence on the ability of RhighresD to capture convective precipitation? Local maxima could be missed due to too coarse station density.

Thank you for pointing this out. We added the discussion of RhiresD quality in the following way on L157: "The dataset suffers from a general tendency to overestimate light rain and underestimate intense rain due to interpolation uncertainty. The uncertainty is higher in data-sparse areas and in cases of high spatial variation (e.g. convective rainfall)".

L153: in-situ observations and remote radar-based observations

Changed.

L167: crowd-sourced reports –> insurance loss reports

Changed.

L169: delete "on the other side" since there is no matching on the one side.

Deleted.

L169: in units of cm starting at 20mm :-)

Changed the unit of hail diameter to mm throughout the manuscript. We corrected this sentence to "MESHS estimates the largest expected hail diameter starting at 20 mm".

L172: 2009 –> 2002

Changed the sentence on L180 to: "C-band radars in operation since 2002 and were later replaced with dual-polarization radars between 2011 and 2012 (Nisi et al., 2018)".

L186: we use data from

Changed.

L200ff: Please explain the calculation of tau in a bit more detail. Is dCAPE/dt a daily mean or daily max value?

The calculation of $\tau$ was described in the L210 in the previous manuscript. We moved this part a couple of lines upward from the previous location to provide the information earlier on L217 in the updated manuscript.

L223: heavy –> severe

Changed.

L255: near the mountainous region of central Europe –> a bit more specific?

Changed to "eastern France and the southern flank of the Alps".

L265: Please explain in a bit more detail how you compare across the different spatial resolutions and how sensitive the method is to the interpolation in space. I suggest adding 1 – 2 sentences of discussion.

For a fair comparison, both observations and simulations are interpolated to the same grid with 12 km grid spacing. This is described in Section 2.5, please see L203. We added the discussion of how sensitive the SAL method is to the interpolation on L210 in the revised manuscript: "The computation requires the identification of rain objects within the analysis domain, separately for the observed and simulated fields. An object is defined as the grid points above the threshold of 1/15 of the maximum

value of rain within the domain following Wernli et al. (2008). As a result, the influence of interpolation on the result is rather small".

L298: Note that the black forest is already quite far away from the radar, and the data quality may become an issue because the radar gates become very big and the lowest beam height is quite high. If you compare only within a 140 km area around the radars, the radar quality should be very good.

We appreciate your input. Indeed, the data quality might be an issue due to beam-widening effects. While there could be a trade-off between the limitation and breadth of the analysis, we believe that the benefits of a broader range could outweigh the potential marginal if we limit the radius within a 140 km radius. Therefore, we suggest maintaining the current figure to allow us to glean insights into model performance to a broader extent.

L303: Please mention that both radar-based hail products only see hail in the air and not on the ground.

Added the sentence on L313: "Furthermore, both MESHS and POH detect hail at higher altitudes and not on the ground, and melting may influence small hailstones (Nisi et al., 2016)".

L371: Please mention the time of the hailfall at the beginning of the case study description.

Thank you for your suggestion. Considering hail was initiated at several locations, we added the time of hail fall from radar-based observations in the description part of the case studies:
L406: "At around 05 UTC, a line of convection developed in northeastern France and moved to the Black Forest (not shown). Hail was first observed in radar-based observations at around 9 UTC over northeastern France and the Jura mountains. Later in the afternoon, a supercell developed near Lyon (in France), moving northeastward and gradually splitting into several elongated convection cells accompanied by observed and simulated long hail swaths extended for hundreds of kilometers over Switzerland (Fig. 10f)".
L425: "This led to a burst of thunderstorms that hit this area in the early morning, where hail was first observed at around 00:30 UTC".
L458: " With significant instability, hail was initially observed at around 12 UTC to the east of the Black Forest when the cold upper-level was advected over the warm near-surface air".
L466: 'Convective cells associated with observed hail and lightning formed on the southern flank of the Jura mountains at around 13 UTC, and later, on the northern flank of the Alps at around 15 UTC".

L380ff: I do not yet fully get my head around the timing of the individual steps and the position of the cold front. You mention the advection of cold air; this would happen behind the cold front. But then you mention the increased dew point ahead of the cold front. It would be great to know where the front is, at which altitude, and at which time to get the full (complex) 4-D picture of the event.

We agree that there might be some confusion with the description. Please find Figure R2 below which shows the 500 hPa geopotential height and 850 hPa temperature for the case 23 July, 2009, from ERA5 reanalysis. We added the description of the cold front on L386: "On that day, central Europe was dominated by a large trough stretched from Scandinavia and its upper low-pressure system positioned north of Iceland. The associated cold front was approached to the north of the Prealps at around 12 UTC".

Regarding the dew-point temperature, we implemented the changes on L396: "Around noon, this profile significantly changed below 600 hPa where temperature decreased due to the approaching of the cold front (Fig. 10e), while dew-point temperatures increased prominently at 800 hPa as the moisture in the warm air mass rises ahead of the cold front and the advection of humid air from the Mediterranean (Fig. 10c)".

L409: becomes warmer –> I do not see this; to me, the temperature near the surface looks colder at 00 UTC than at 12 UTC.

Thank you for pointing this out. We implemented the changes on L423: "At 00 UTC on 25 June 2017, the air below 900 hPa became cooler and the lid was higher, indicating that the capping inversion

[Figure]

**Figure R2:** Synoptic situation for 23 July 2009. Panels show geopotential height at 500 hPa (m, black contours), temperature at 850 hPa (°C, shaded) obtained from ERA5 reanalysis.

was penetrated by updrafts. Together with the potential instability ahead of an upper-level trough, the buoyant air was lifted and released the accumulated energy. This led to a burst of thunderstorms that hit this area in the early morning, where hail was first observed at around 0030 UTC".

L418: Given the southeasterly flow –> this implies causality; please explain in more detail.

We implemented the changes on L434: "As shown in Fig. 2b, due to the presence of an upper-level trough, the Po Valley was influenced by the southwesterly flow. A line of [...]".

L432: Please explain in a bit more detail why the stable was eroded by the upper-level westerly flow.

We implemented the changes on L448: "The stable layer was eroded due to the warming of the near-surface air in the morning hours, making the conditions [...]".

L439: add not shown

We deleted this irrelevant sentence, and updated the associated description on L257.

L461: It is unclear to what "this" refers and what is meant by these kind of events (I think there is some grammatical confusion as to whether you mean the synoptically driven cases or not)

We implemented the changes on L478: "This is to some extent associated with the chaotic nature of the underlying dynamics and the lower predictability of the localized events".

L471: three cases that were selected

Changed.

L485: In this context... this statement is not very clear to me yet

This sentence has been deleted as it does not fit in this context.

**Figures**

Fig.1: caption four –> three Changed.

Fig.11: text says the sounding is at Payerne, but the figure title suggests the sounding is at Stuttgart.

Changed. We apologize for this confusion, and the text should be Stuttgart. We can see the stratospheric intrusion better at Stuttgart compared to Payerne, because it's closer to the hailstorm.

Fig.11: Please add the borders of Switzerland.

We have the borders of Switzerland in Fig. 9-11. Do you mean to make the borders thicker?

**References**

Adams-Selin, R. D. (2023). A three-dimensional hail trajectory clustering technique. *Monthly Weather Review*, 151(9):2361–2375.

Fierro, A. O., Mansell, E. R., MacGorman, D. R., and Ziegler, C. L. (2013). The implementation of an explicit charging and discharge lightning scheme within the WRF-ARW model: Benchmark simulations of a continental squall line, a tropical cyclone, and a winter storm. *Monthly Weather Review*, 141(7):2390–2415.

Nisi, L., Hering, A., Germann, U., and Martius, O. (2018). A 15-year hail streak climatology for the Alpine region. *Quarterly Journal of the Royal Meteorological Society*, 144(714):1429–1449.

Nisi, L., Martius, O., Hering, A., Kunz, M., and Germann, U. (2016). Spatial and temporal distribution of hailstorms in the Alpine region: A long-term, high resolution, radar-based analysis. *Quarterly Journal of the Royal Meteorological Society*, 142(697):1590–1604.

Wernli, H., Paulat, M., Hagen, M., and Frei, C. (2008). SAL - A novel quality measure for the verification of quantitative precipitation forecasts. *Monthly Weather Review*, 136(11):4470–4487.

**Response to Reviewer 2**

**Review of manuscript WCD-2023-11 "Evaluation of hail and lightning climatology using km-scale climate model over the Alpine region"**

by Ruoyi Cui, Nikolina Ban, Marie-Estelle Demory, Raffael Aellig, Oliver Fuhrer, Jonas Jucker, Xavier Lapillonne, Christoph Schär

The authors compare 2.2 km COSMO simulations driven by reanalysis data with different observations of total precipitation, hail, and lightning of a number of severe weather events. Overall, the paper is written sufficiently clearly, and I only have a (relatively large number of) somewhat minor comments.

We thank anonymous Reviewer 2 for his/her valuable comments and positive feedback on our study. Below are the replies to his/her comments in blue.

**Major points**

It is not quite clear what the goal of the paper is: Assess the COSMO diagnostics or to explain why severe convective weather occurred on those days. I have to admit that I didn't get much out of the short case studies (which are partially redundant with the case summaries earlier in the paper); is there anything new to be learned from these?

We thank the reviewer for the comments. We agree that the short case studies might not fully address the second objective on L84. We try to update the three listed objectives to two: (1) Evaluate the performance of the COSMO model at km-scale grid spacing in simulating hail and lightning; (2) Explore how the COSMO model represents storm environments through case studies.

Throughout the manuscript, there are statements touting the simulations as performing "very well." I think such statements are subjective and should be avoided unless a metric has been defined and justified, which quantifies the performance of the simulations. I suggest deleting these statements and simply report the differences between the observations and simulations, and let the reader decide to what extent the model diagnostics are useful (or maybe offer your take on the results in the conclusions section without overselling the quality of the simulations).

We thank the reviewer for the feedback. We appreciate the reviewer's perspective regarding the statements that may appear subjective, illustrating the simulations as performing "very well". To address this concern, we revised the manuscript by removing the subjective statements and focusing on reporting the difference between observations and simulations. We have reviewed the examples provided in the minor points section and addressed each one individually, please have a look.

**Minor points**

L4ff: Since you are still in the km scale, the simulations are convection allowing, but not convection resolving (to resolve convective flows, a grid spacing of O(100) m is required).

We changed the sentence to "These kilometer-scale models improve [...]". Later, use the km-scale models throughout the manuscript.

L8: Write out the COSMO acronym before using it the first time.

We implemented the changes on L7: "The simulations are performed with the climate version of the regional model Consortium for Small-scale Modeling (COSMO) that runs on Graphics Processing Units (GPUs) at a horizontal grid spacing of 2.2 km".

L13: Is the topographic barrier a requirement, or is it just present? This feature does not seem to be highlighted later on, and I'm not sure it has been demonstrated that this barrier was indeed a cause of the severe weather. More analysis/discussion would be needed to grant this feature a place in the abstract.

The topographic barrier mentioned here does play a crucial role in shaping the local atmospheric conditions for severe weather, for example, the 25 June 2017 morning case over the Po Valley. We added a sentence on L439 to highlight this topographic feature: "The topography of the Po Valley offers a favorable environment for the initiation of new cells, which consequently explains the hotspot of hail occurrence over this region".

L25: Reword thunderstorm peril region (e.g., regions at highest risk of experiencing thunderstorms).

We implemented the changes on L24: "It is recognized as one of the regions at high risk of experiencing thunderstorms in Europe due to its notable topography and proximity to the Mediterranean Sea".

L44: Can you be more specific about which aspects are not well-understood? Such broad statements tend to dismiss the wealth of research that has been done in this area.

We changed the first sentence on L44 to be more specific: "Although severe convective storms can cause catastrophic damage, important processes such as hail growth pathways (Adams-Selin, 2023) and charging physics of the cloud (Fierro et al., 2013)) for predicting hail and lightning are insufficiently represented in weather and climate models".

L47: Wind shear also promotes the organization of quasi-linear systems and updraft intensification (e.g., Markowski and Richardson 2010).

We changed the sentence and added reference on L48 as suggested: "[...], and wind shear that can promote the storm organization and intensification of updrafts (e.g., Markowski and Richardson (2010))".

L49: One would expect a given environment to lead to the same storm, whether or not the storm is surrounded by mountains. Can you specify what you mean by stating that the role of the ingredients is complicated? Do you mean that there is a larger suspected variability of the thermodynamics/kinematic fields compared to flat terrain, or that the environment cannot easily be sampled (because it may be rather inhomogeneous)? Or do you specifically talk about the lift ingredient (the subsequent text seems to imply such)?

We agree the statement is quite vague. We changed the sentence on L51 to "Furthermore, over complex topography, additional thermodynamic and kinematic mechanisms may also affect the initiation and development of convection. For example, [...]".

L53: Grammar broken (. . . and subsequently to. . . )

We implemented the changes on L53: "High insolation during the day contributed to large latent heat fluxes, resulting in moisture accumulation within the valley. This moisture was subsequently transported to the mountain crest via upvalley winds".

L54: "The" mesoscale boundary hasn't been introduced yet (could say "a" mesoscale convergence zone).

Changed "the mesoscale convergence zone" into "a mesoscale convergence zone".

L102: Isn't 1 h is the interval (rather than a frequency)?

Changed "updating frequency" into "updating interval".

L103: Add AGL/MSL after 23.5 km

We implemented the changes on L104: "[...], where vertical spacing ranges from 20 m near the surface to 1.2 km at the model top located at around 23.5 km above mean sea level".

L114: Calculated instead of accumulated?

Updraft duration is calculated, we updated the explanation on L114 to make it clearer: "If the grid column or any adjacent grid columns has a maximum updraft exceeding $10\,\mathrm{m\,s^{-1}}$ between the previous and current model time steps, the updraft duration field is incremented by one model time step. This field is used to track the convective cells and limit the maximum updraft time in the hail model".

L123: Are there also non-convective thunderstorms?

Actually, there are also non-convective thunderstorms (e.g. spider lightning, dry lightning, etc.), mostly cloud-to-ground lightning. As this statement was taken from Lynn and Yair (2010) and LPI is used for convective thunderstorms (via the non-inductive mechanism), we have decided to keep the sentence as it is. The case study from Yair et al. (2010) illustrates that WRF-calculated LPI might not predict lightning caused by stratiform precipitation very well.

L140: Maybe add that these values are much lower than what is observed in the real world because of the relatively coarse grid spacing (the convective clouds tend to be wider, and updrafts weaker than in truly convection resolving simulations).

Thanks for pointing it out. We added the sentence as you suggested on L143: "Note that these values are much lower than what is observed in the real world due to the simulated wider convective clouds and weaker updrafts, given the 2.2 km grid spacing is high but not to a level that matches reality".

L141: Why is CAPE negative?

We corrected the unit of this buoyancy — the value $-1500\,\mathrm{J\,kg^{-2}}$ here is the threshold for the filter $f_2$ discussed in the appendix B (Equation 16) in Brisson et al. (2021). The equation is formally similar to mixed-layer CAPE but with a different integration layer. It is used to remove the spurious LPI signals under the conditions of gravity waves embedded in moist flow (e.g. during Föhn events in the Alps in winter). It is approximately 0 or slightly negative at locations of explicitly simulated convective cells, but attains large negative values for the stable conditions associated with orography mountain waves. We added: "see Eq. 16 in Brisson et al. (2021)" on L145 to provide detailed information for this integrated buoyancy.

L147: Delete spatial

Changed.

L157: Please add the size in cm for each category.

Added the size in mm accordingly on L164-168, and decided to change the units to mm throughout the manuscript to be consistent.

L164: Hailpad observations.

Changed.

L180: LINET data are not gridded (right?), so they can't really have a resolution. Perhaps say that the location error is 3 km? Why is the time known only to within 2 min? Given that the system has something like nano-second accuracy internally, 2 min seems like a huge inaccuracy.

LINET are not gridded, in fact, it has an average location accuracy of approximately 150 m. We modified the sentence on L188 to "LINET has an average location accuracy of approximately 150 m (Betz et al., 2009). The LINET data is taken from ?, the total lightning is gridded at 3 km grid spacing with a temporal resolution of 2 min. Higher temporal resolution and spatial resolutions are possible, but due to high computational and storage demands, we use this 2D database in this study. Nonetheless, it

still provides sufficient information to discern local characteristics. Later, we aggregated the LINET lightning flashes every hour to compare against the simulated hourly maximum LPI".

L192: Write out the acronym before using the abbreviated form.

We implemented the changes on L204: "We then use the SAL (Structure-Amplitude-Location) method [...]".

L198: Reword: e.g., ...indicates that the model field is too spread-out/broad/diffuse. Also recommend avoiding these parenthetical constructions to shorten the sentence.

Shortened the sentence on L204 to "The A component is calculated as the normalized difference between the domain-averaged observed and simulated fields. A positive (negative) A component indicates an overestimation (underestimation) by the model. The L component considers the displacement of the center of mass between the observed and simulated fields, as well as the weighted average distance between individual objects and the mass center of the total field. Lower L values indicate a more accurate placement of the simulated field. The S component accounts for the size and shape of the objects. Positive (negative) S values suggest a more widespread (peaked) simulated field".

L225: "This cold front" hasn't been introduced yet

Thank you for your comment. We changed "this cold front" to "a cold front", and later added the description of the front's location in section 3.4.1 on L386: "On that day, central Europe was dominated by a large trough stretched from Scandinavia and its upper low-pressure system positioned north of Iceland. The associated cold front was approached to the north of the Alps at around 12 UTC".

L229: Please specify what was discharged (water?)

Changed the text to "[...] caused water discharges with return periods of 10 to 30 years [...]".

L234: damage (instead of damages)

Changed.

L265, 274: Model showing good performance: This seems like an entirely subjective statement. Either define what you consider as "good" or leave that judgment to the reader (and just report the errors).

We rewrote the text to illustrate the performance instead of making subjective statements:
L265 now on L275: The SAL diagram of daily accumulated precipitation is shown in Fig. 3.
L274 now on L284: Comparison against IMERG and high-resolution RhiresD observations reveals that COSMO can capture the main spatial distributions of daily accumulated rainfall (Fig.4).

L267: Reword: . . . captured relatively well

Changed.

L288: Model reproducing hail "very well": Maybe the general presence of hail is predicted well, but the placement and coverage is (not surprisingly) not captured very well (e.g., 8 July 2017). Like above, I suggest reporting the differences and omit statements about the quality of the simulations.

We implemented the changes on L288: "In general, the occurrence of hail is simulated well, but the placement and coverage are not captured very well in some cases (e.g., 8 July 2017)".

L301: . . . hailstone diameters above 20 mm.

Changed.

L315: Another instance of "good" model performance.

We implemented the changes on L326: "Moreover, when compared to hailpad observations over Croatia (panels d-f), the model exhibits a distribution that closely aligns with the observed data, particularly

for the two cases of 25 June 2017 and 17 May 2018. For the case of 24 July 2017, an event relatively well captured by the model, the simulated hailstones above 15 mm are underestimated".

L321: Suggest replacing precipitation with rain or "total precipitation" (hail is also "precipitation").

We replaced the precipitation with total precipitation, and used rain instead of precipitation for the case of 1 June 2013 (heavy rainfall without hail).

L325: It is not surprising that the model is unable to produce accurate hail sizes; the updrafts are barely resolved, you are using single-moment microphysics, etc. I would probably consider omitting the hail size comparison, or at least state upfront that the model cannot be expected to produce accurate hail sizes (forecasting hail size accurately remains a holy grail of severe weather prediction).

We agree that the model cannot be expected to produce accurate hail sizes. However, it could be useful to show the comparison against available observations. As raised by Reviewer 1, it's unfair to compare the frequency from HAILCAST and MESHS, because MESHS only provides an estimation of hail size >20 mm. Instead, we changed the panel (a-h) in Fig.6 by comparing the area affected by hail from observations and the model. The updated panels (a-h) are shown in Figure R1. As the panels (a-h) present different comparisons (e.g., area and frequency), we broke the old fig.6 into two figures.

L375: If the situation is strongly synoptically-forced, then presumably there is a good amount of flow in the troposphere, typically leading to fast storm motions. Why do the storms become quasi-stationary?

We agree that this could lead to confusion, and the sentences are updated starting from L386: "On that day, central Europe was dominated by a large trough stretched from Scandinavia and its upper low-pressure system positioned north of Iceland. The associated cold front was approached to the north of the Prealps at around 12 UTC. The propagation of the front was slow due to the distortion of the flow field around the Alps (Schumann, 1987), while the convergence along the front resulted in a fast storm movement. According to Schemm et al. (2016), up to 45% of detected hail events in north-eastern and southern Switzerland form in this kind of pre-frontal zone".

L444: True, but updrafts also suffer more from entrainment, so the net effect tends to be less evaporative cooling (James and Markowski, 2010, MWR).

Thank you for this comment and the reference. We implemented the changes on L460: "Due to the dry and cold air aloft, evaporative cooling and melting of hydrometeors could lead to stronger and colder downdrafts (Johns and Doswell, 1992). While updrafts also experience entrainment, the overall effect tends to be less evaporative cooling (James and Markowski, 2010)".

**Figures**

Fig.4: Consider adding labels to the rows to make it easier to identify which panels refer to the observations, and which to the simulations.

Thank you for your suggestions. Fig.4 is updated and we added labels to the rows to make it more reader-friendly.

Fig.5: The panels, and especially the legends are too small; consider breaking this figure into two.

Thank you for your suggestions. Fig.5 is updated and we moved the legends outside the figures and adjusted the size of the legends.

**References**

Adams-Selin, R. D. (2023). A three-dimensional hail trajectory clustering technique. *Monthly Weather Review*, 151(9):2361–2375.

Betz, H. D., Schmidt, K., Laroche, P., Blanchet, P., Oettinger, W. P., Defer, E., Dziewit, Z., and Konarski, J. (2009). LINET-An international lightning detection network in Europe. *Atmospheric Research*, 91(2-4):564–573.

Brisson, E., Blahak, U., Lucas-Picher, P., Purr, C., and Ahrens, B. (2021). Contrasting lightning projection using the lightning potential index adapted in a convection-permitting regional climate model. *Climate Dynamics*, 57(7-8):2037–2051.

Fierro, A. O., Mansell, E. R., MacGorman, D. R., and Ziegler, C. L. (2013). The implementation of an explicit charging and discharge lightning scheme within the WRF-ARW model: Benchmark simulations of a continental squall line, a tropical cyclone, and a winter storm. *Monthly Weather Review*, 141(7):2390–2415.

James, R. P. and Markowski, P. M. (2010). A numerical investigation of the effects of dry air aloft on deep convection. *Monthly Weather Review*, 138(1):140–161.

Johns, R. H. and Doswell, C. A. (1992). Severe local storms forecasting. *Weather and Forecasting*, 7(4):588–612.

Lynn, B. and Yair, Y. (2010). Prediction of lightning flash density with the WRF model. *Advances in Geosciences*, 23(8):11–16.

Markowski, P. and Richardson, Y. (2010). *Mesoscale Meteorology in Midlatitudes*. Wiley.

Schemm, S., Nisi, L., Martinov, A., Leuenberger, D., and Martius, O. (2016). On the link between cold fronts and hail in Switzerland. *Atmospheric Science Letters*, 17(5):315–325.

Schumann, U. (1987). Influence of mesoscale orography on idealized cold fronts. *Journal of the Atmospheric Sciences*, 44(23):3423–3441.

Yair, Y., Lynn, B., Price, C., Kotroni, V., Lagouvardos, K., Morin, E., Mugnai, A., and del Carmen Llasat, M. (2010). Predicting the potential for lightning activity in mediterranean storms based on the weather research and forecasting (WRF) model dynamic and microphysical fields. *Journal of Geophysical Research*, 115(D4).